# Feature-Mapping Topology Optimization with Neural Heaviside Signed Distance Functions

Aleksandr Kolomeitsev [1]    Anh-Huy Phan [1]

## Abstract

Topology optimization plays a crucial role in designing efficient and manufacturable structures. Traditional methods often yield free-form voids that, although providing design flexibility, introduce significant manufacturing challenges and require extensive post-processing. Conversely, feature-mapping topology optimization reduces post-processing efforts by constructing topologies using predefined geometric features. Nevertheless, existing approaches are significantly constrained by the limited set of geometric features available, the variety of parameters that each type of geometric feature can possess, and the necessity of employing differentiable signed distance functions. In this paper, we present a novel method that combines Neural Heaviside Signed Distance Functions (Heaviside SDFs) with structured latent shape representations to generate manufacturable voids directly within the optimization framework. Our architecture incorporates encoder and decoder networks to effectively approximate the Heaviside function and facilitate optimization within a unified latent space, thus addressing the feature diversity limitations of current feature-mapping techniques. Experimental results validate the effectiveness of our approach in balancing structural compliance, offering a new pathway to CAD-integrated design with minimal human intervention.

## 1. Introduction

Accelerating the development of manufacturable structures with minimal human input is a significant concern in the design industry. While tools like Computer-Aided Design (CAD) have reduced early design errors and expedited the creation of documentation, there remains a need for enhanced automation, particularly in design decision-making.

Component design relies on meeting strength characteristic requirements and manufacturability criteria, meaning that components must be tailored to specific, predefined manufacturing methods.

Topological optimization allocates material within a design space to meet strength or thermal requirements. However, it encounters challenges, including complex geometries that necessitate advanced manufacturing techniques and extensive post-processing, which can diminish the time advantages gained from minimal engineer involvement in the topology optimization process.

Recent advancements have addressed manufacturability concerns in topology optimization. In additive manufacturing, manufacturability-focused approaches, such as stress-minimization topology optimization, tackle practical issues by incorporating connectivity constraints into the optimization process (Chao et al., 2021). Additionally, projection-based algorithms enforce nozzle size restrictions in material extrusion-based additive manufacturing, ensuring compliance with production constraints (Carstensen, 2020).

Innovative computational techniques, including performance-aware diffusion models, surpass GAN-based methods in manufacturability-aware topology optimization by integrating constraint guidance into the optimization process (Mazé & Ahmed, 2023). Furthermore, machine learning approaches automate the adjustment of geometric constraints and optimize building orientations for additive manufacturing, thereby enhancing the feasibility of topology-optimized structures (Mohseni & Khodaygan, 2024). Recent studies, such as (Zehnder et al., 2021), indicate that neural topology optimization with implicit representations and self-supervised learning can match the performance of state-of-the-art mesh-based solvers.

Multi-scale design methods further enhance manufacturability by bridging the gap between macro- and micro-level design features, facilitating manufacturable topology optimization while reducing computational costs (Groen, 2018;

[1] Artificial Intelligence Center, Laboratory of Intelligent Signal and Image Processing, Skolkovo Institute of Science and Technology, Moscow, Russia. Correspondence to: Aleksandr Kolomeitsev <aleksandr.kolomeitsev@skoltech.ru>.

*Proceedings of the 42nd International Conference on Machine Learning*, Vancouver, Canada. PMLR 267, 2025. Copyright 2025 by the author(s).

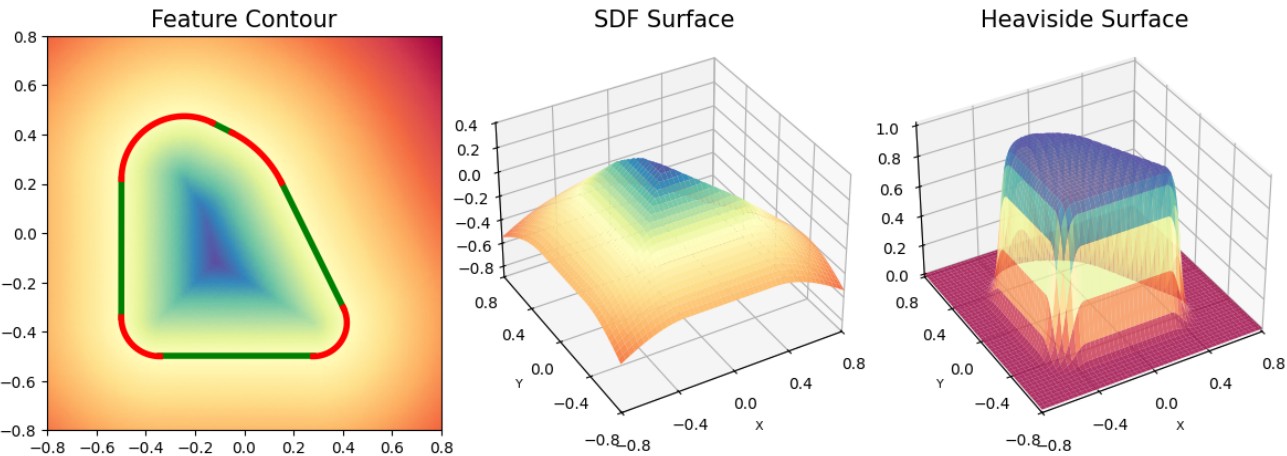

*Figure 1.* Example of SDF and Heaviside function for quadrangle geometric feature

Xia & Breitkopf, 2015; Zhu et al., 2017). Additionally, techniques like advection-diffusion-based filtering ensure machinable designs by applying efficient computational filters to unstructured meshes (Høghøj & Träff, 2022).

Feature-based modeling incorporates geometric primitives to produce CAD-compatible designs during topology optimization, thereby facilitating direct manufacturability (Liu & Ma, 2015). Moreover, constructive solid geometry (CSG) approaches optimize structural boundaries using implicit representations, enhancing scalability and control over complex geometries (Mei et al., 2008).

Thus, topology optimization is increasingly transitioning from a conceptual tool to a practical enabler of manufacturable designs that align with real-world constraints.

Building on this potential, our research aims to integrate advanced optimization techniques with machine learning paradigms to overcome the inherent limitations of current feature-mapping methods in topological optimization. By addressing these challenges, we strive to enhance the scalability, flexibility, and overall effectiveness of topology optimization processes, ultimately leading to the development of more efficient and manufacturable structural designs.

To generate manufacturable solutions through topology optimization that replicate those created by engineers using CAD, geometric features must consist of closed chains of linear and arc segments connected tangentially, without restricting the number of elements in the chain. However, existing feature-mapping methods cannot accommodate such geometric representations. Although it is possible to explicitly express the Signed Distance Function (SDF) (see Fig. 1) within these methods, calculating gradients at segment contacts presents significant challenges. Moreover, these methods lack the ability to seamlessly transform one type of geometric feature into another.

In response to these challenges, we propose leveraging an variational autoencoder-based model to approximate the Heaviside function, thereby providing a unified representation for diverse geometric features within a single latent space. We illustrate this approach using examples of ellipses, triangles, and quadrilaterals. Furthermore, we demonstrate how the decoder of the trained model can be utilized for topology optimization by treating the latent representation of a geometric feature as its parameters. The code and models are publicly available at https://github.com/Alexander19970212/NHSDF-TOp.

## 2. Related Work

Topology optimization involves the optimization of material distribution within a design domain to achieve maximum structural performance under specified constraints. It has been applied across various fields, including structural engineering (Izumi et al., 2024), fluid dynamics (Li et al., 2023), and thermal systems (Yu et al., 2020).

**SIMP.** One of the most widely utilized methods in topology optimization is the Solid Isotropic Material with Penalization (SIMP) method (Bendsøe & Kikuchi, 1988), which employs a continuous density variable (or pseudo-density) to interpolate material properties between solid and void states, where values close to zero correspond to the void state and values near one represent the solid state. Typically, each pseudo-density variable is associated with a specific element within the finite element method (FEM) discretized domain (see Appendix B). The objective function is commonly formulated to minimize compliance while adhering to a volume fraction constraint. Two of the most prevalent approaches for solving the optimization problem formulated in the SIMP framework are the Optimality Criteria (OC) method and the Method of Moving Asymptotes (MMA), both of which utilize adjoint differentiation for

sensitivity analysis (Bendsøe & Sigmund, 2004).

**Feature Mapping Methods**. Feature mapping methods optimize the sizes and positions of geometric features within the design domain to ensure that FEM elements inside the feature are designated as either material (Norato et al., 2015) or voids (Saxena, 2010; Zhang et al., 2017a; Wein et al., 2020). Various studies have represented geometric features using hyperellipses (Guo et al., 2014; Zhang et al., 2017b; Sharma et al., 2017; Norato, 2018), B-splines (Lee & Kwak, 2008; Lee et al., 2007; Kim et al., 2008), and Bézier curves (Wang & Yang, 2009). In a recent work (Padhy et al., 2025), the authors introduce polygon-based geometric features constructed by combining half-spaces.

To achieve a smooth transition of pseudo-density at the feature boundaries, the Heaviside Signed Distance Function (SDF) is utilized. The SDF value for a point is defined as the distance to the feature boundary, with a positive sign if the point resides inside the feature and a negative sign otherwise. This SDF value is transformed into a pseudo-density value through a Heaviside function, which can be implemented using hyperbolic tangent function (Wein & Stingl, 2018), polynomials (Zhang et al., 2016; Dunning, 2018), or trigonometric functions (Norato et al., 2015). Many feature mapping methods incorporate various smooth combination functions to ensure continuous derivatives with respect to high-level parameters, such as the Kreisselmeier-Steinhauser function (Shapiro, 2002) or smooth R-functions (Chen et al., 2007).

By maintaining smoothness throughout the computational process SDF-based methods can leverage a range of gradient-based optimization techniques (Zhang & Norato, 2018). In these approaches, derivatives with respect to parameters are calculated using the chain rule, analogous to the techniques used in SIMP with the adjoint method (Tröltzsch, 2010). It is noteworthy that some proposed methods employ a non-differentiable maximum function, necessitating the use of stochastic optimization methods such as genetic algorithms (Wang & Yang, 2009), evolutionary strategies (Bujny et al., 2017), Bayesian optimization (Sharpe et al., 2018), and support vector regression (Lei et al., 2018) as alternative solutions.

**Neural Network Approximation of the Heaviside SDF**. Neural networks have been employed to approximate the Heaviside function for boundary transitions in parametric level-set methods for topology optimization (Deng & To, 2021). Beyond topology optimization, neural networks are utilized to approximate SDFs in various applications. A prominent example is DeepSDF (Park et al., 2019), which encodes 3D shapes and represents continuous signed distance fields. MetaSDF (Sitzmann et al., 2020) enhances SDF representation by leveraging previously learned knowledge. Additionally, (Chibane et al., 2020) introduces Neural Distance Fields to represent distance fields for non-closed surfaces, while (Jiang et al., 2020) employs neural networks to represent SDFs with differentiability for solving inverse problems such as 3D reconstruction.

## 3. Background

The table of all notations used in this paper is shown in Appendix A.

### 3.1. FEM Discretization for 2D Elasticity Problem

The finite element method (FEM) is employed to discretize the design domain for the 2D elasticity problem by dividing it into a mesh of finite elements, typically bilinear quadrilaterals or triangular elements. Each element is associated with a set of nodal degrees of freedom corresponding to displacements in the $x$ and $y$ directions. The equilibrium equations for the elasticity problem are derived using the principle of virtual work, leading to the following system of linear equations

$$\mathbf{K}\mathbf{u} = \mathbf{f} \tag{1}$$

where $\mathbf{K}$ is the global stiffness matrix, $\mathbf{u}$ is the displacement vector, and $\mathbf{f}$ is the external force vector. The stiffness matrix $\mathbf{K}$ is assembled from the elemental stiffness matrices, which are functions of the material properties and the geometry of each element. As an example, the assembling of the stiffness matrix is shown in Appendix B.

### 3.2. SIMP Method Problem Formulation

The Solid Isotropic Material with Penalization (SIMP) method is a widely used approach in topology optimization for determining the optimal material distribution within a design domain. The primary objective is to minimize the compliance $C$ of the structure, defined as the work done by the external forces

$$\min_{\rho} \quad C(\rho) = \mathbf{u}^T \mathbf{f} \tag{2}$$

subject to the equilibrium equation

$$\mathbf{K}(\rho)\mathbf{u} = \mathbf{f} \tag{3}$$

and a volume constraint

$$\sum_{e=1}^{N} v_e \rho_e \leq V_{\max} \tag{4}$$

where $\rho_e \in [\rho_{\min}, 1]$ represents the pseudo-density variable for element $e$, and $v_e$ is the volume of element $e$. $N$ is the total number of elements, and $V_{\max}$ is the maximum allowable volume, $\rho_{\min} \geq 0$ is the minimum allowable pseudo-density. The stiffness matrix $\mathbf{K}(\rho)$ is interpolated using the

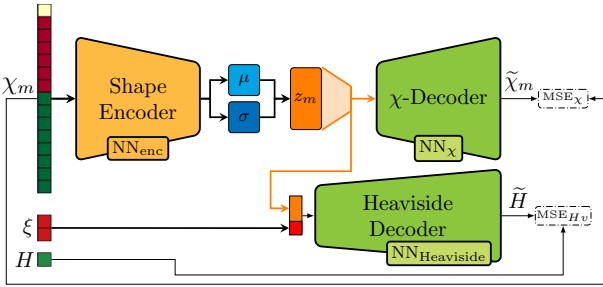

*Figure 2.* Neural Heaviside SDF Model

SIMP penalization scheme

$$\mathbf{K}(\rho) = \sum_{e=1}^{N} \rho_e^p \mathbf{K}_e \tag{5}$$

where $\mathbf{K}_e$ is the element stiffness matrix of the solid material of the element $e$ and $p > 1$ is the penalization factor that encourages the solution towards a 0-1 (black-and-white) design.

### 3.3. Feature-Mapping Topology Optimization (FMTO) Problem Formulation

Feature-Mapping Topology Optimization (FMTO) extends traditional topology optimization by incorporating geometric features into the design process. In FMTO, pseudo-densities are parameterized using the defining attributes of these geometric features, such as scale, offset, and rotation angle. The optimization problem can thus be formulated as

$$\begin{aligned} \min_s \quad & C(s) = \mathbf{u}^T \mathbf{K}(\rho(s))\mathbf{u} = \sum_{e=1}^{N} \rho_e^p(s)\mathbf{u}^T \mathbf{K}_e \mathbf{u} \\ \text{s.t.} \quad & \mathbf{K}(\rho(s))\mathbf{u} = \mathbf{f} \\ & \sum_{e=1}^{N} v_e \rho_e(s) \leq V_{\max} \end{aligned} \tag{6}$$

where $s$ represents the set of feature parameters. The feature mapping involves defining a correspondence between the design variables $s$ and the element pseudo-densities $\rho$, typically through a Signed Distance Function (SDF) representation. This mapping ensures that the geometric features are smoothly integrated into the design domain, facilitating a more controlled and manufacturable optimization process.

## 4. Method

### 4.1. Neural Signed Distance Function

In the proposed method, we use a neural approximation of the Heaviside function of the Signed Distance Function (SDF) of geometric features to represent voids in Feature Mapping Topology Optimization.

**SDF and Heaviside Function.** For a specified geometric feature $m$ we define a subset of parameters $s_m$, the signed distance function (SDF) is defined as

$$sdf(s_m, \xi) = \begin{cases} d(\xi, \partial\Omega(s_m)) & \text{if } \xi \in \Omega(s_m) \\ -d(\xi, \partial\Omega(s_m)) & \text{otherwise} \end{cases} \tag{7}$$

where $d(\xi, \partial\Omega(s_m))$ represents the distance from point $\xi$ to the boundary of feature $m$. $\Omega(s_m)$ is the domain occupied by the geometric feature, parameterized by $s_m$.

The motivation for using the Heaviside function is to scale the distance function to the range $[0, 1]$, which is convenient for the optimization process and for training the neural network. We approximate the Heaviside function using a sigmoid function

$$H(s_m, \xi) = \frac{1}{1 + e^{-\beta\, sdf(s_m, \xi)}} \tag{8}$$

where $\beta$ controls the steepness of the sigmoid function (see Fig. 1), typical value is $\beta = 20$.

By employing a neural approximation, we obtain a differentiable approximation of the Heaviside function in a new latent space, such that $\widetilde{H}(s_m, \xi) \approx \mathrm{NN}_{\text{Heaviside}}(z_m, \xi)$, where $z_m$ is a latent representation of the feature parameters $s_m$ mapped using shape encoder.

**Neural Heaviside SDF.** The proposed model is based on variational autoencoder-based architecture (Kingma & Welling, 2022) consisting of a shape encoder and several decoders (see Fig. 2). During training, we establish a common latent space for different geometric features by minimizing multiple loss functions.

**Shape Code.** The shape code $\chi_m$ is a vector of 15 elements organized in a one-hot manner: for each feature type, a specific segment of the code is activated while the remaining elements are set to zero, except for the first element $\chi_m[0]$, which serves as a normalized label. For instance, in our experiments, an ellipse is labeled as 0, a triangle as 0.5, and a quadrangle as 1. For an ellipse, $\chi_m[1]$ parameterizes the semi-axis ratio, with the major axis fixed at 0.25. In the case of a triangle, $\chi_m[2, 3]$ specify the coordinates of its movable vertex, while the other vertices are fixed at $(-0.5, -0.5)$ and $(0.5, -0.5)$. The elements $\chi_m[4:6]$ represent the rounded radii of the triangle's vertices. For a quadrangle, $\chi_m[7, 8]$ and $\chi_m[9, 10]$ denote the coordinates of the two top vertices, respectively, while the bottom vertices remain fixed as in the triangle. Finally, $\chi_m[11:14]$ specify the rounded radii of the quadrangle's vertices. This scheme is illustrated in Figure 3.

**Shape Encoder.** The encoder receives a shape code $\chi_m$ for a geometric feature $m$, and outputs a latent vector, $z_m = \mathrm{NN}_{\text{enc}}(\chi_m)$.

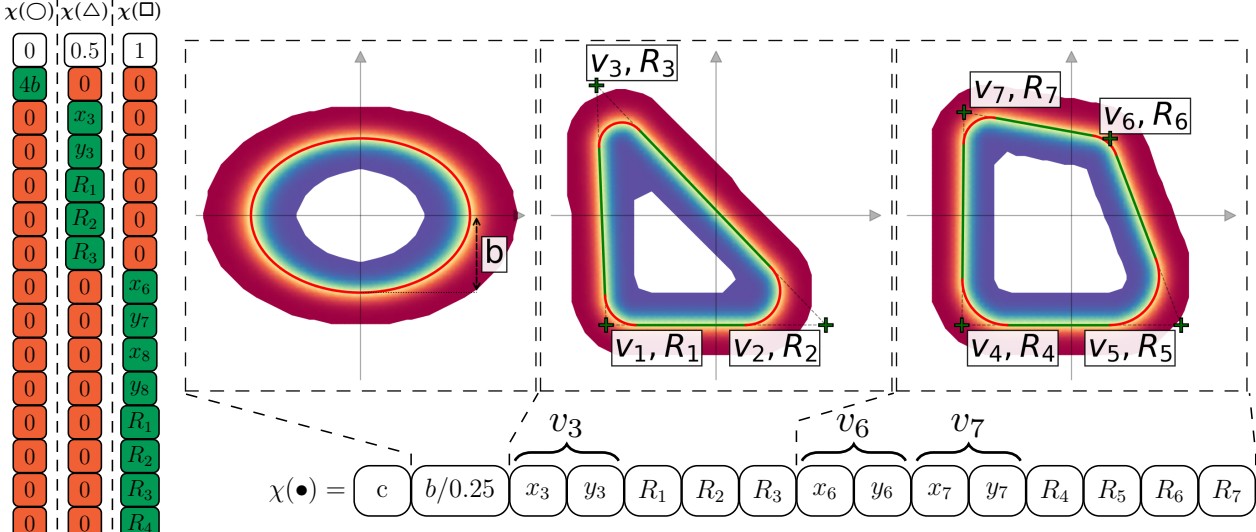

*Figure 3.* Assembly of Shape Code $\chi$ for Various Geometric Features. Here, **c** represents a normalized label, **b** denotes the minor axis of an ellipse, $v_i$ and $R_i$ indicate the vertices and rounded radii of the i-th polygon corner, and $x_i$ and $y_i$ are the coordinates of the i-th vertex of the polygon.

**Heaviside Decoder.** The decoder takes the latent vector $z_m$ concatenated with the point coordinates $\xi$ as an input, and outputs the value of $\widetilde{H}$ at that point, $\widetilde{H} = \text{NN}_{\text{Heaviside}}(z_m, \xi)$.

**Reconstruction Decoder.** This decoder is responsible for reconstructing the shape code from the latent space, $\widetilde{\chi}_m = \text{NN}_{\text{recon}}(z_m)$.

**Loss Functions.** During training, we minimize the mean squared error (MSE) between the predicted and true values of $H$ and $\chi_m$. The Kullback-Leibler divergence ($D_{\text{KL}}$) is employed as a regularization term to ensure that the latent distribution is close to the Gaussian distribution.

The total loss function is defined as

$$
L = \frac{1}{B} \sum_{i=1}^{B} \left( H(\chi_i, \xi_i) - \text{NN}_{\text{Heaviside}}(\text{NN}_{\text{enc}}(\chi_i), \xi_i) \right)^2 \\
+ \frac{\lambda_{\text{KL}}}{B} \sum_{i=1}^{B} D_{\text{KL}}(q_\phi(z_i|\chi_i)||p(z_i))
\tag{9}
$$

where $B$ is the batch size, $\lambda_{\text{KL}}$ is the regularization parameter for the KL divergence, $q_\phi(z_i|\chi_i)$ is the encoder distribution for the latent vector $z_i$ given the shape code $\chi_i$, and $p(z_i)$ is the prior (normal) distribution over the latent vector $z_i$.

It is important to note that the reconstruction decoder is trained separately, while all other parts of the model remain frozen. The MSE loss is exclusively applied to the reconstruction decoder

$$
L_{\text{recon}} = \frac{1}{B} \sum_{i=1}^{B} \left( \chi_i - \text{NN}_{\text{recon}}(\text{NN}_{\text{enc}}(\chi_i)) \right)^2
\tag{10}
$$

## 4.2. Feature Mapping Topology Optimization with Neural Heaviside SDF. Inference

During inference, we employ the pretrained Heaviside Decoder with fixed weights to approximate the Heaviside function for the geometric features represented by the latent vectors. This approximation also enables us to compute the gradients required to adjust these latent vectors with respect to the compliance objective function.

**Problem Formulation.** Unlike the classical approach to topology optimization, which specifies the final volume requirement as a constraint, our proposed method incorporates this requirement into the objective function. This requirement becomes active only if the volume exceeds a specified threshold. Apart from this constraint, the problem formulation remains similar to classical FMTO, where the goal is to minimize compliance, and the optimization is performed with respect to high-level geometric feature variables that define the boundaries of voids

$$
\begin{aligned}
\min_{\{w,b,\alpha,z\}} \quad & \sum_{e=1}^{N} \rho_e^p \mathbf{u}^T \mathbf{K}_e \mathbf{u} + \lambda_{\text{vol}} \max(0, v^T \rho - V_{\max}) \\
\text{s.t.} \quad & \rho_e = 1 - \max_{m=1}^{M} \left( \text{NN}_{\text{Heaviside}}(\hat{\xi}_{e,m}, z_m) \right) \\
& \hat{\xi}_{e,m} = w_m(\xi_e - b_m)R(\alpha_m) \\
& \mathbf{K}(\rho)\mathbf{u} = \mathbf{f}
\end{aligned}
\tag{11}
$$

where $M$ is the number of geometric features, $w_m$ and $b_m$ are the scale and offset of feature $m$, $R(\alpha_m)$ is the rotation matrix, $\alpha_m$ is the rotation angle, and $z_m$ represents the shape variables for feature $m$ in the latent space. Consequently, $s_m$ is defined as $s_m = \{w_m, b_m, \alpha_m, z_m\}$. $\xi_e$ is the centroid coordinates of the element $e$.

The motivation behind formulating the pseudo-density function is to obtain a combined Heaviside function that yields a value close to 1 when a point lies within one of the features and close to 0 otherwise, thereby creating a void within the feature.

To achieve a smooth approximation of the maximum function, we employ the Kreisselmeier-Steinhauser function, taking into account the $\rho$ limits

$$
\max_{m=1...M} (\mathrm{NN}_{Heaviside}(\hat{\xi}_{e,m}, z_m)) \approx
$$
$$
\approx \frac{1 - \rho_{\min}}{\gamma_{\mathrm{KS}}} \ln \left( \sum_{i=1}^{M} \exp(\gamma_{\mathrm{KS}} \mathrm{NN}_{Heaviside}(\hat{\xi}_{e,m}, z_m)) \right)
$$
$$(12)$$

where $\gamma_{\mathrm{KS}}$ is the smoothing parameter.

**Gradient Chain Rule for High-Level Design Variables.** To solve the optimization problem with the combined objective function, we employ an iterative method using adjoint differentiation. The sensitivity of the function $J(s_m)$ with respect to a high-level design variable $s_m$ is computed as follows

$$
\frac{\partial J}{\partial s_m} = \sum_{e=1}^{N} \left[ \left( \frac{\partial J}{\partial \rho_e} + p\,\rho_e^{p-1} \mathbf{u}^T \mathbf{K}_e \mathbf{u} \right) \frac{\partial \rho_e}{\partial s_m} \right] \quad (13)
$$

## 5. Implementation Details

**Variable Initialization.** Variables are initialized such that, in the first iteration, geometric features form a regular grid of squares with maximum rounded radii. The initial value of $z_m$ for the first iteration is computed via Shape Encoder.

**Variable Limitations.** To constrain the values of design variables, preventing them from becoming too small or too large and ensuring they remain within the design domain, we apply sigmoid reparameterization

$$
s_{m,j} = (s_{\max,j} - s_{\min,j}) \frac{1}{1 + \exp(-\hat{s}_{m,j})} + s_{\min,j} \quad (14)
$$

where $s_{\max,j}$ and $s_{\min,j}$ are the maximum and minimum allowable values of the design variable $s_{m,j}$, respectively; $j = 1, 2, 3, 4$.

As a result, we minimize the objective function with respect to the logits of the design variables $\hat{s}_{m,j}$.

**Refactoring Mechanism.** To prevent the latent variable to be infeasible, we realize refactoring mechanism, where every 5 iterations we update the value of $z_m$ using the reconstruction Decoder and Encoder

$$
z_m = \mathrm{NN}_{\mathrm{enc}}(\mathrm{NN}_{\mathrm{recon}}(z_m^{\mathrm{old}})) \quad (15)
$$

Additional implementation details include the implementation of the KS function and the selection of its parameters. Further details are provided in Appendix G.

## 6. Results

### 6.1. Neural Heaviside SDF

To evaluate the performance of the Neural Heaviside SDF, we conducted several experiments using generated datasets and a comprehensive set of metrics.

**Metrics.** The accuracy of the predictions for $H$ and $\chi$ was assessed using Mean Squared Error (MSE). For the topology optimization process to converge effectively, it is crucial to minimize noise in the predicted $H$. To evaluate this noise, we compute the norm of the gradient of the predicted $H$ using the finite difference method on grid points

$$
\mathrm{Smth} = \frac{1}{N_p} \sum_{i=1}^{N_p} \| \nabla \left( \mathrm{NN}_{Heaviside}(z_m, \xi_i) - H(\chi_m, \xi_i) \right) \|_2
$$
$$(16)$$

where $N_p$ represents the number of sensor points. This metric was computed across each $m$-th feature in the test dataset and subsequently averaged.

**Datasets.** Our datasets include training and testing datasets for the Heaviside Decoder and the Reconstruction Decoder. The dataset for training the Heaviside Decoder contains samples of geometric features and randomly located points with their corresponding Heaviside values. The dataset for training the Reconstruction Decoder contains only samples of shape codes. To evaluate the **Smth** metric, we generated a dataset that includes feature code shapes and a grid of points with their Heaviside values. Further details can be found in Appendix E.

**Model Architecture.** The architecture of the Encoder consists of a block of fully-connected layers with batch normalization and LeakyReLU activation functions. The architecture of the Reconstruction Decoder mirrors that of the Encoder. The Heaviside Decoder utilizes the architecture of the DeepSDF decoder (Park et al., 2019). For more details on the model architecture, please refer to the Appendix C. To demonstrate the effectiveness of using the DeepSDF model as a decoder, we trained various model types, including Autoencoders (AE), Variational Autoencoders (VAE),

and Maximum Mean Discrepancy VAEs (MMD-VAE), ensuring that each model had an equivalent number of parameters in their respective blocks. A symmetric decoder model mirroring the encoder was also employed for comparison with the DeepSDF Decoder.

The comparative results are presented in Table 1 for 20 model training runs. Additionally, we conducted a t-test to confirm the superiority of the best combination of training strategies and model architectures over the other combinations. The results are presented in Appendix F.

*Table 1.* Comparison of different model architectures with and without DeepSDF as decoder and different training strategies.

| | | $\mathrm{MSE_{Hv}}$ | $\mathrm{MSE_{\chi}}$ | **Smoothness** |
|---|---|---|---|---|
| **Training latent via Heaviside Dec.** | **DeepSDF Decoder** | | | |
| | AE | 0.000346 ± 1.81e-05 | 0.0002 ± 3.4e-05 | 0.00586 ± 9.95e-05 |
| | MMD-VAE | 0.000368 ± 1.95e-05 | 0.000228 ± 3.52e-05 | 0.006 ± 0.000102 |
| | VAE | **0.000277** ± **1.07e-05** | 0.000202 ± 2.82e-05 | **0.00575** ± **7.11e-05** |
| | **Symmetric Decoder** | | | |
| | AE | 0.0014 ± 0.000268 | 0.000229 ± 3.5e-05 | 0.0165 ± 0.000658 |
| | MMD-VAE | 0.00134 ± 0.000233 | 0.000212 ± 3.08e-05 | 0.0161 ± 0.000667 |
| | VAE | 0.00128 ± 0.000201 | 0.000117 ± 2e-05 | 0.0163 ± 0.000574 |
| **Training latent via Reconstr. Dec.** | **DeepSDF Decoder** | | | |
| | AE | 0.000796 ± 0.000101 | 0.000226 ± 5.04e-05 | 0.0084 ± 0.000286 |
| | MMD-VAE | 0.00059 ± 5.42e-05 | 0.000195 ± 1.61e-05 | 0.00818 ± 0.000403 |
| | VAE | 0.000475 ± 3.51e-05 | 7.42e-05 ± 1.09e-05 | 0.00806 ± 0.000187 |
| | **Symmetric Decoder** | | | |
| | AE | 0.00155 ± 0.000257 | 0.000232 ± 5.27e-05 | 0.0175 ± 0.00044 |
| | MMD-VAE | 0.0015 ± 0.00023 | 0.000198 ± 2.21e-05 | 0.017 ± 0.000704 |
| | VAE | 0.00146 ± 0.000327 | **7.02e-05** ± **4.82e-06** | 0.017 ± 0.000888 |

In all cases, we observe improved metrics when adding DeepSDF compared to the symmetric decoder.

**Training Strategies.** We also compared two training strategies: **1**. Training the latent representation using the Heaviside decoder and then fine-tuning the Reconstruction decoder on the trained latent representation. **2**. Training the latent representation using the Reconstruction decoder

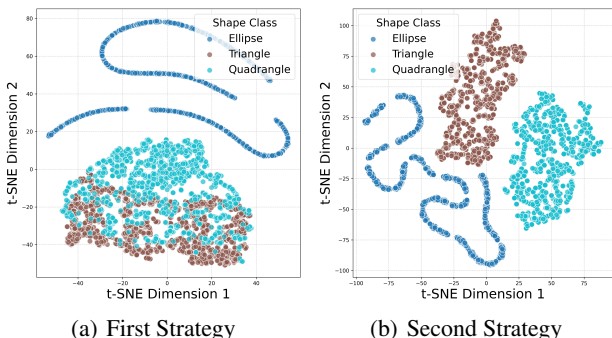

(a) First Strategy      (b) Second Strategy

*Figure 4.* Comparison of latent representations of geometric features clusters for different training strategies. **a)** First strategy. Training latent representation using the Heaviside decoder. **b)** Second strategy. Training latent representation using the Reconstruction decoder.

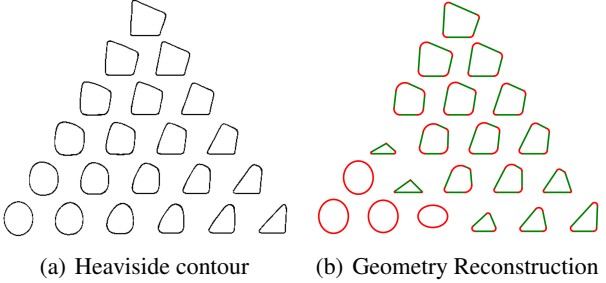

(a) Heaviside contour      (b) Geometry Reconstruction

*Figure 5.* Reconstruction of geometric features from simplex transition between features in the latent space. **a)** Reconstruction of contour from Heaviside values. **b)** Reconstruction of geometry through the Reconstruction decoder.

and then fine-tuning the Heaviside decoder on the trained latent representation.

Notably, the first strategy achieves good metrics for both $\mathrm{MSE_{Hv}}$ and $\mathrm{MSE_{\chi}}$. In contrast, the second strategy yields better metrics for $\mathrm{MSE_{\chi}}$ but does not achieve sufficiently good metrics for $\mathrm{MSE_{Hv}}$. We suggest that training the latent representation with the Heaviside decoder results in a latent space tailored specifically to the contours of geometric features. Conversely, training with the Reconstruction decoder focuses the latent representation on the shape code of the features without adequately considering their geometry. This is indirectly supported by the distribution of latent vectors reduced to a 2D plane using t-SNE (Laurens van der Maaten & Hinton, 2008), as shown in Figure 4.

Based on these presumptions, we can consider a triangle as a special case of a quadrangle from the perspective of the Heaviside contour, where one vertex connects collinear edges. This results in overlapping distribution clusters for these features (see Figure 6.1). Additionally, for ellipses, the transitional shapes to other features in our dataset are not well-represented, leading to noticeably poorer transition quality. Figure 5 clearly shows the transitions in latent

space between different geometric features using the Heaviside decoder and the Reconstruction decoder.

## 6.2. Topology Optimization

We present the results of utilizing the Neural Heaviside SDF for feature mapping in topology optimization across several classic cases. Detailed descriptions and examples of solutions obtained using the SIMP method are provided in Figure 6 and Appendix D.

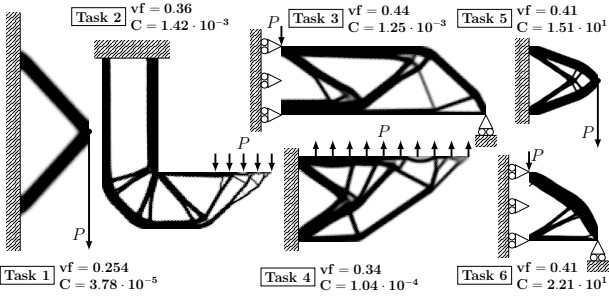

*Figure 6.* Classical topology optimization tasks, and solutions obtained using the SIMP method.

*Table 2.* Comparison of different topology optimization methods via **Compliance metric**, the final volume fraction (vf) is given in brackets.

| | Method | Task 1 $\times 10^{-5}$ | Task 2 $\times 10^{-3}$ | Task 3 $\times 10^{-3}$ | Task 4 $\times 10^{-4}$ | Task 5 $\times 10^{1}$ | Task 6 $\times 10^{1}$ |
|---|---|---|---|---|---|---|---|
| Free-Form | SIMP | **3.78** (0.254) | **1.42** (0.36) | **1.25** (0.44) | **1.04** (0.34) | **1.51** (0.41) | **2.21** (0.41) |
| | TopoDiff | N/A | N/A | N/A | N/A | 1.70 (0.424) | 2.44 (0.415) |
| | NTopo | 4.08 (0.255) | N/A | 1.63 (0.438) | 1.07 (0.339) | 1.60 (0.412) | 2.31 (0.41) |
| FMTO | Ellipses | 4.39 (0.297) | 2.22 (0.368) | 1.74 (0.449) | 1.55 (0.345) | 1.58 (0.426) | 2.45 (0.416) |
| | TreeTOp | 5.85 (0.266) | N/A | 3.73 (0.455) | 1.62 (0.356) | 1.57 (0.449) | 3.09 (0.439) |
| | NHSDF (Ours) | 3.91 (0.255) | 1.98 (0.354) | 1.63 (0.437) | 1.52 (0.337) | 1.55 (0.409) | 2.34 (0.401) |

Among the various Feature Mapping methods, we implemented an approach similar to (Kumar & Saxena, 2022), where all geometric features are represented by ellipses. Additionally, we use the official TreeTOp implementation (Padhy et al., 2025) for comparison, where geometric features are represented as polygons constructed from a set of half-spaces. Utilizing ellipses as geometric features significantly limits the value of the parameter $V_{max}$. In our experiments, we used the minimum volume fraction (vf) required to obtain a valid solution. Here, the volume fraction is defined as the ratio of the volume of solid elements to the total volume of the design domain. Specifically, with our method, we intentionally reduced the target value of $V_{max}$

to demonstrate that even with less material, better compliance values can be achieved. Refer to the comparison in Table 2.

Additionally, we present comparative results from NTopo (Zehnder et al., 2021) and TopoDiff (Mazé & Ahmed, 2023). Both methods are free-form topology optimization approaches that use neural models to predict optimal topology solutions. NTopo employs a deep learning technique to parameterize the density and displacement fields within the optimization process. TopoDiff uses a conditional diffusion model as a surrogate guidance strategy that actively favors designs with low compliance. The TopoDiff method is implemented only for square design domains. Therefore, we introduced two new tasks (Tasks 5 and 6), which replicate Tasks 1 and 3 on square domains. Although NTopo supports domain modifications, defining boundary conditions is challenging, so we did not apply it to these tasks. TreeTOp is implemented exclusively for rectangular domains. Consequently, the comparison for the Bracket problem (Task 2) includes only the SIMP and Ellipses FMTO approaches. The final topology results for all methods are presented in Figure 7, and the von Mises stress distribution plots are provided in Appendix H.

For FMTO methods, the metric values obtained by SIMP and NTopo are unattainable because they can create small, locally conditioned edges through free-form approach, which our methods counteract to ensure manufacturability. So SIMP method as added to estimate the lower bound of the compliance metric. However, it is worth noting that our method closely matches the SIMP solution when the topology is simpler, as observed in Task 1.

Table 2 shows that among all FMTO methods (TreeTOp, Ellipses, and NeuralHeavisideSDF), our approach achieves the best compliance metric while using less material. Additionally, in some experiments our method is comparable to NTopo (see Task 3) and outperforms TopoDiff on square-domain tasks.

## 7. Limitations and Future Work

In the previous section, we demonstrated successful integration of Neural Heaviside Signed Distance Function for Feature Mapping topology optimization. Nevertheless, several limitations must be addressed, which also offer avenues for future research.

**Poor transitions between different geometric feature types.** We observed that shape transitions are largely corrupted by the Reconstruction decoder, whereas the Heaviside decoder produces accurate contours for intermediate shapes.

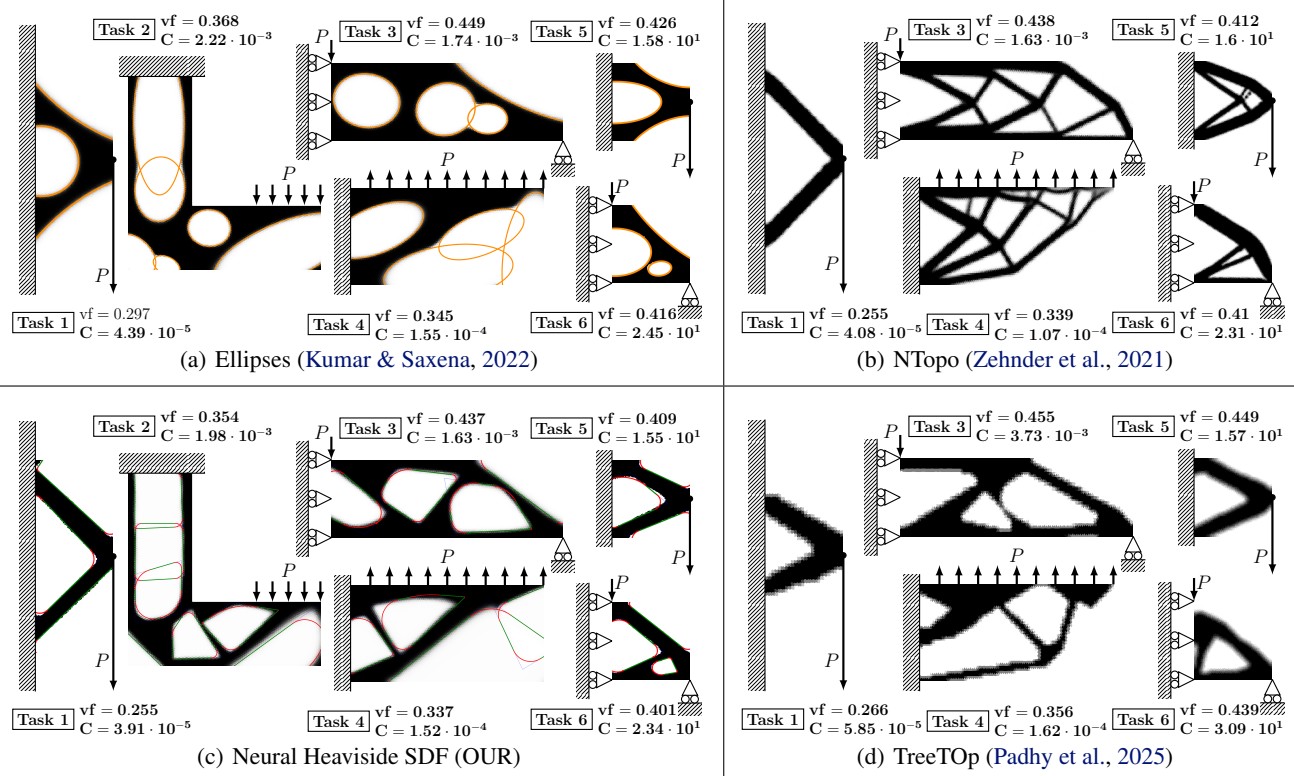

*Figure 7.* Classical topology optimization tasks and corresponding solutions obtained using different methods: (a) Ellipses, (b) NTopo, (c) Neural Heaviside SDF, and (d) TreeTOp.

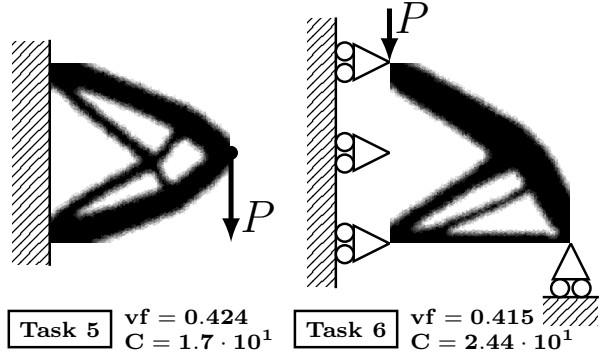

*Figure 8.* Solutions obtained using the TopoDiff method (Mazé & Ahmed, 2023) for Square domain tasks.

**Presence of a large number of conflicting variables during optimization.** This issue necessitates alternating the optimized variables or periodically decreasing their learning rate. Furthermore, the optimization process struggles to stabilize the offset variables, leading to oscillatory movements of geometric figures at the end of the optimization. Currently, this problem is effectively resolved by artificially slowing down the optimization speed.

As noted in (Wein et al., 2020), and as we also emphasize, topology optimization outcomes depend on initial conditions. Future work will explore improved initialization strategies and expand our geometric feature set.

We performed a t-test to assess the statistical significance of our model approximator's performance relative to other models, and we present the results in Appendix F. In a future journal paper, we will provide a detailed analysis of our model's robustness.

## 8. Conclusion

In this work, we have demonstrated several new approaches that can significantly expand the capabilities of FM topology optimization: generalization of parameters of various geometric features through latent space; a mechanism for training the latent representation via a Heaviside decoder, which allows focusing on the geometry of features; a refactoring mechanism that avoids the issue of the latent representation exceeding the permissible domain and enables control over individual high-level feature parameters. Our experiments demonstrate that this approach can be applied to Feature Mapping topology optimization, thereby ensuring the technological feasibility of the optimized solution, as FM approaches are always technologically feasible. We anticipate that the proposed approaches can be extended to more complex tasks, involving a greater variety of geometric features and three-dimensional (3D) problems.

## Impact Statement

This paper presents research aimed at advancing the field of Feature Mapping Topology Optimization (FMTO) through the application of neural network methodologies. While there are several potential societal implications of our work, we do not consider any of them to warrant detailed discussion within the scope of this manuscript.

## Acknowledgments

The work was supported by the grant for research centers in the field of AI provided by the Ministry of Economic Development of the Russian Federation in accordance with the agreement 000000C313925P4F0002 and the agreement with Skoltech №139-10-2025-033.

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

# A. Notations and Definitions

*Table 3.* Notations and Definitions (Part I)

| Notation | Definition |
| --- | --- |
| $\mathbf{K}$ | Global stiffness matrix obtained by assembling all element stiffness matrices $\mathbf{K}_e$. |
| $\mathbf{K}_e$ | Element stiffness matrix for element $e$. The size of the matrix is equal to the size of matrix $\mathbf{K}$. |
| $k_e$ | Local element stiffness matrix (for element $e$) before assembly. It is equavelent to the matrix $\mathbf{K}_e$ but without the zero rows and columns. |
| $\mathbf{u}$ | Nodal displacement vector in the FEM discretization. |
| $\mathbf{f}$ | External force vector. |
| $e$ | Index of element in the FEM mesh. |
| $N$ | Total number of elements in the FEM mesh. |
| $\rho_e$ | Pseudo-density variable for element $e$ with $\rho_e \in [\rho_{\min}, 1]$. |
| $\rho$ | Vector of pseudo-density variables for all elements in the FEM mesh. |
| $\rho_{\min}$ | Minimum allowable pseudo-density (to avoid singular stiffness matrices). |
| $v_e$ | Volume of element $e$. |
| $v$ | Vector of volumes for all elements in the FEM mesh. |
| $V_{\max}$ | Maximum allowable volume in the design (volume constraint). |
| $p$ | Penalization factor in the SIMP interpolation ($\rho_e^p$), with $p > 1$. |
| $C(\rho)$ or $C(s)$ | Compliance of the structure, defined as $\mathbf{u}^T\mathbf{f}$ or $\mathbf{u}^T\mathbf{K}\mathbf{u}$. |
| $s$ | Set of high-level design variables defined the set of geometric features. |
| $s_m$ | Subset of parameters defining geometric feature $m$. |
| $M$ | Total number of geometric features used in the optimization. |
| $sdf(s_m, \xi)$ | Signed distance function for geometric feature $m$ at point $\xi$. |
| $\xi$ | A point in the design domain. |
| $\Omega(s_m)$ | Domain occupied by geometric feature $m$ as determined by $s_m$. |
| $\partial\Omega(s_m)$ | Boundary of the geometric feature $m$. |
| $d(\xi, \partial\Omega(s_m))$ | Euclidean distance from point $\xi$ to the boundary $\partial\Omega(s_m)$. |
| $H(s_m, \xi)$ | Approximated Heaviside of the signed distance function for feature $m$ at point $\xi$ using a sigmoid function. |
| $\beta$ | Steepness parameter in the sigmoid approximation of the Heaviside function. Typical value is $\beta = 20$. |
| $\widetilde{H}(s_m, \xi)$ | Neural approximation of the Heaviside of the signed distance function for feature $m$ at point $\xi$ using latent representation of the feature parameters $s_m$ mapped using shape encoder. |
| $\mathrm{NN}_{\mathrm{Heaviside}}(z_m, \xi)$ | Neural network approximator for the Heaviside of the signed distance function for feature $m$ at point $\xi$, taking latent vector $z_m$ as input. |
| $z_m$ | Latent vector associated with geometric feature $m$, obtained via the shape encoder. |
| $\chi_m$ | Shape code for geometric feature $m$ (the encoded representation of its parameters). |
| $\mathrm{NN}_{\mathrm{enc}}$ | Shape Encoder network that maps $\chi_m$ to the latent vector $z_m$. |
| $\widetilde{\chi}_m$ | Reconstructed shape code for geometric feature $m$ using the reconstruction decoder. |
| $\mathrm{NN}_{\mathrm{recon}}$ | Reconstruction Decoder that maps the latent vector $z_m$ back to a reconstructed shape code $\chi_m$. |

*Table 4.* Notations and Definitions (Part II)

| Notation | Definition |
|---|---|
| $B$ | Batch size used during training (in the loss computations). |
| $\chi$ | Batch of shape codes for all geometric features. |
| $z$ | Batch of latent vectors associated with all geometric features, obtained via the shape encoder. |
| $D_{\mathrm{KL}}$ | Kullback-Leibler divergence function. |
| $\lambda_{\mathrm{KL}}$ | Regularization parameter for the KL divergence. |
| $q_\phi(z_i|\chi_i)$ | Encoder distribution for the latent vector $z_i$ given the shape code $\chi_i$. |
| $p(z_i)$ | Prior distribution over the latent vector $z_i$. |
| $\lambda_{\mathrm{vol}}$ | Penalty factor for volume constraint violations in the objective function. |
| $\xi_e$ | A center point of element $e$. |
| $\hat{\xi}_{e,m}$ | Transformed coordinate for point $\xi_e$ with respect to feature $m$; defined as $$\hat{\xi}_{e,m} = w_m(\xi_e - b_m)R(\alpha_m).$$ |
| $w_m$ | Scale factor for geometric feature $m$ used to adjust its size in topology optimization. |
| $b_m$ | Offset (translation) for geometric feature $m$. |
| $\alpha_m$ | Rotation angle for geometric feature $m$. |
| $R(\alpha_m)$ | Rotation matrix corresponding to the angle $\alpha_m$. |
| $\gamma_{\mathrm{KS}}$ | Smoothing parameter in the Kreisselmeier-Steinhauser function used for smooth maximum approximation. Typical value is $\gamma_{\mathrm{KS}} = 40$. |
| $s_{\min,j},\ s_{\max,j}$ | Lower and upper bounds for the design variables $s_{m,j}$. Since for the proposed method, the $s_m$ is defined as $s_m = \{w_m, b_m, \alpha_m, z_m\}$, so $s_{\min,1} = w_{\min}$, $s_{\min,2} = b_{\min}$, $s_{\min,3} = \alpha_{\min}$, and $s_{\min,4} = z_{\min}$ and similarly for the upper bounds. |
| $\hat{s}_{m,j}$ | Unconstrained (logit) representation of design variable $s_{m,j}$ before sigmoid reparameterization. |
| Smth | Smoothness metric quantifying the noise in the neural Heaviside approximation: $$\mathrm{Smth} = \frac{1}{N_p} \sum_{i=1}^{N_p} \left\| \nabla\Big(\mathrm{NN}_{\mathrm{Heaviside}}(z_m, \xi_i) - H(\chi_m, \xi_i)\Big) \right\|_2.$$ |
| $N_p$ | Number of sensor points used to compute the smoothness metric. |
| $N_{\mathrm{node}}$ | Total number of nodes in the FEM mesh. |
| $\lambda, \mu$ | Lamé constants used in the elasticity (Hooke) matrix construction. |
| $\mathrm{dofs}_{\mathrm{Free}}$ | Set of degrees of freedom with unknown displacements (free DOFs). |
| $\mathrm{dofs}_{\mathrm{FixedMoved}}$ | Set of degrees of freedom with prescribed (fixed) displacements. |

## B. Assembling of the Global Stiffness Matrix

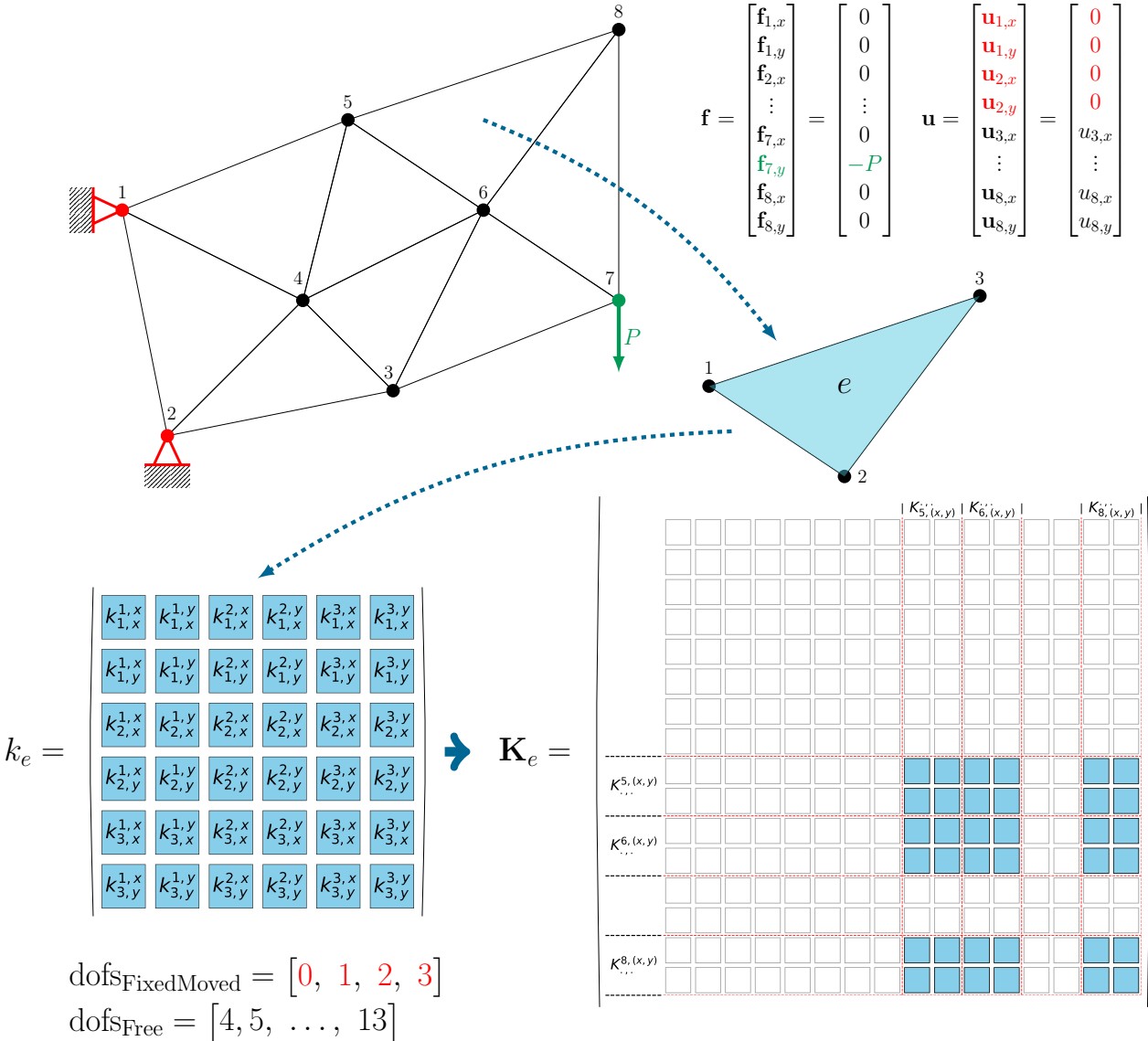

*Figure 9.* Example of building the element stiffness matrix for a triangular element for a domain discretized with 8 nodes in the FEM method.

In this section, we do not provide the full derivation of the system of linear equations from the 2D elasticity differential equations. Instead, we present only assembling of the element stiffness matrix. Comprehensive explanations can be found in (Cenanovic & Jonsson).

The discretized displacement field for the domain $\Omega$ with $N_{\text{node}}$ nodes is represented by the vector

$$\mathbf{u} = \left[\mathbf{u}_{1,x}, \mathbf{u}_{1,y}, \mathbf{u}_{2,x}, \mathbf{u}_{2,y}, \ldots, \mathbf{u}_{i,x}, \mathbf{u}_{i,y}, \ldots, \mathbf{u}_{N_{\text{node}},x}, \mathbf{u}_{N_{\text{node}},y}\right]^T \tag{17}$$

where $\mathbf{u}_{i,x}$ and $\mathbf{u}_{i,y}$ are the displacement components along the $x$ and $y$ axes, respectively, and $i$ denotes the node index.

Consider a local element $e$ within the discretized domain, for which a local stiffness matrix $k_e$ is constructed using the Galerkin method and Mendel notation (Galerkin, 1915). For a triangular element, the matrix dimensions are 3dof $\times$ 3dof, where dof represents the number of degrees of freedom per node. In the 2D case, dof $= 2$, making $k_e$ a $6 \times 6$ matrix. From

this matrix, an element stiffness matrix $\mathbf{K}_e \in \mathbb{R}^{2N_{node} \times 2N_{node}}$ is constructed such that all values are zero except for those corresponding to the degrees of freedom of the nodes belonging to element $e$. An example of assembling the stiffness matrix for a domain discretized with 8 nodes can be found in Figure 9.

To assemble the global stiffness matrix $\mathbf{K} \in \mathbb{R}^{2N_{node} \times 2N_{node}}$, it is necessary to sum all the element stiffness matrices $\mathbf{K}_e$

$$\mathbf{K} = \sum_{e=1}^{N} \mathbf{K}_e \tag{18}$$

During the topology optimization process, pseudo-density variables $\rho_e$ are used to indicate the solidity of each element

$$\mathbf{K} = \sum_{e=1}^{N} \rho_e^p \mathbf{K}_e \tag{19}$$

$$0 < \rho_{\min} \leq \rho_e \leq \rho_{\max}$$

**Implementation of boundary conditions**. Let $\text{dofs}_{\text{FixedMoved}}$ denote the set of degrees of freedom with known displacements, and $\text{dofs}_{\text{Free}}$ denote the set of degrees of freedom with unknown displacements (see Fig. 9). The global stiffness matrix used for the solution is obtained by removing the rows and columns corresponding to $\text{dofs}_{\text{FixedMoved}}$ and incorporating the known displacements into the right-hand side of the system of equations

$$\mathbf{K}[\text{dofs}_{\text{Free}}, \text{dofs}_{\text{Free}}] \cdot \mathbf{u}[\text{dofs}_{\text{Free}}] = \mathbf{f}[\text{dofs}_{\text{Free}}] - \mathbf{K}[\text{dofs}_{\text{Free}}, \text{dofs}_{\text{FixedMoved}}] \cdot \mathbf{u}[\text{dofs}_{\text{FixedMoved}}] \tag{20}$$

During optimization, whenever an update to $\mathbf{u}$ is required, the system in Equation 20 is solved, and the entire vector $\mathbf{u}$ is updated with the solution values.

## C. Model Architecture

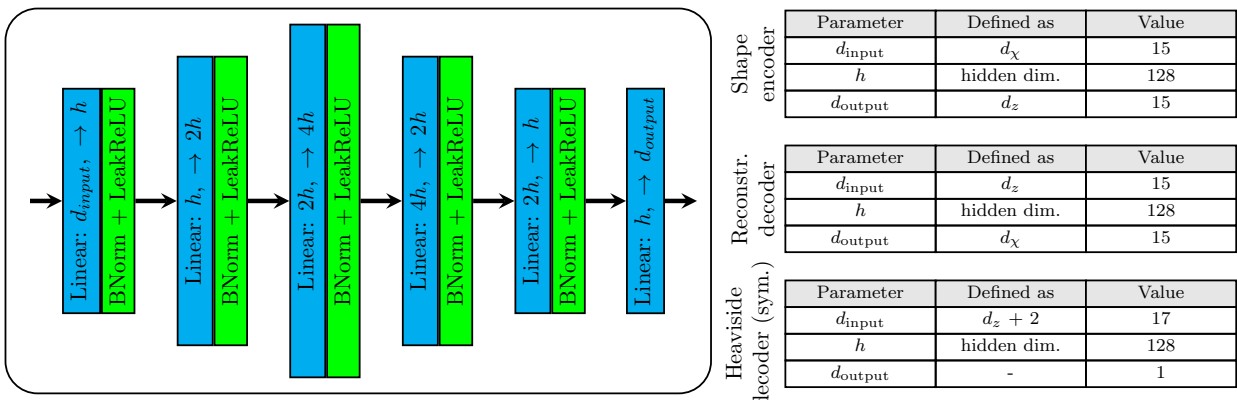

| Shape encoder | Parameter | Defined as | Value |
|---|---|---|---|
| | $d_{\text{input}}$ | $d_\chi$ | 15 |
| | $h$ | hidden dim. | 128 |
| | $d_{\text{output}}$ | $d_z$ | 15 |

| Reconstr. decoder | Parameter | Defined as | Value |
|---|---|---|---|
| | $d_{\text{input}}$ | $d_z$ | 15 |
| | $h$ | hidden dim. | 128 |
| | $d_{\text{output}}$ | $d_\chi$ | 15 |

| Heaviside decoder (sym.) | Parameter | Defined as | Value |
|---|---|---|---|
| | $d_{\text{input}}$ | $d_z + 2$ | 17 |
| | $h$ | hidden dim. | 128 |
| | $d_{\text{output}}$ | - | 1 |

(a) Architecture used for the Encoder, Reconstruction Decoder, and symmetric version of the Heaviside Decoder

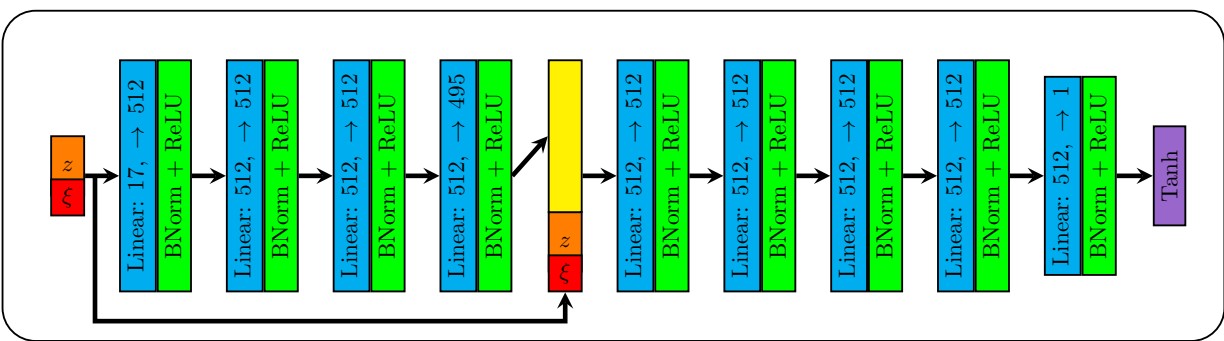

(b) Architecture of the Heaviside Decoder, using the DeepSDF architecture

*Figure 10.* Schematics and parameters of the model's architecture.

Figure 10(a) shows the architecture used for the Encoder, Reconstruction Decoder, and the symmetric version of the Heaviside Decoder. The Encoder architecture consists of a block of fully-connected layers with batch normalization and LeakyReLU activation functions. The Reconstruction Decoder mirrors the Encoder's architecture. Figure 10(b) illustrates the architecture of the Heaviside Decoder, which utilizes the DeepSDF decoder architecture (Park et al., 2019).

## D. Topology Optimization Schemes

The detailed schemes of the tasks are presented in Figure 11.

**Task 1: Cantilever Beam**. The design domain is fixed on the left side, while a load of $P = -0.0025$ N is applied at the midpoint of the right side. The rectangular domain measures $0.25$ m $\times$ $0.60$ m.

**Task 2: Bracket**. This task has an L-shaped form with dimensions $0.15$ m $\times$ $0.15$ m. The top part is fixed, and a uniform load of $P = -0.0025$ N is distributed on the top part of the protrusion.

**Task 3: MBB Beam Half Design Domain**. This domain typically represents a symmetric half of the MBB beam. The design domain measures $1.2$ m $\times$ $0.4$ m. A vertical rolling condition is applied to the left side, and a horizontal rolling condition is applied to the bottom right corner. A load of $P = -0.0025$ N is applied at the upper left corner.

**Task 4: Cantilever Beam with distributed load**. The domain of the task is a rectangle with dimensions $1$ m $\times$ $0.50$ m. A distributed load of $P = 0.0025$ N is applied to the upper part of the domain, and the left side is fixed.

**Task 5: Square Cantilever Beam**. This task replicates Task 1 with the design domain modified to a square measuring

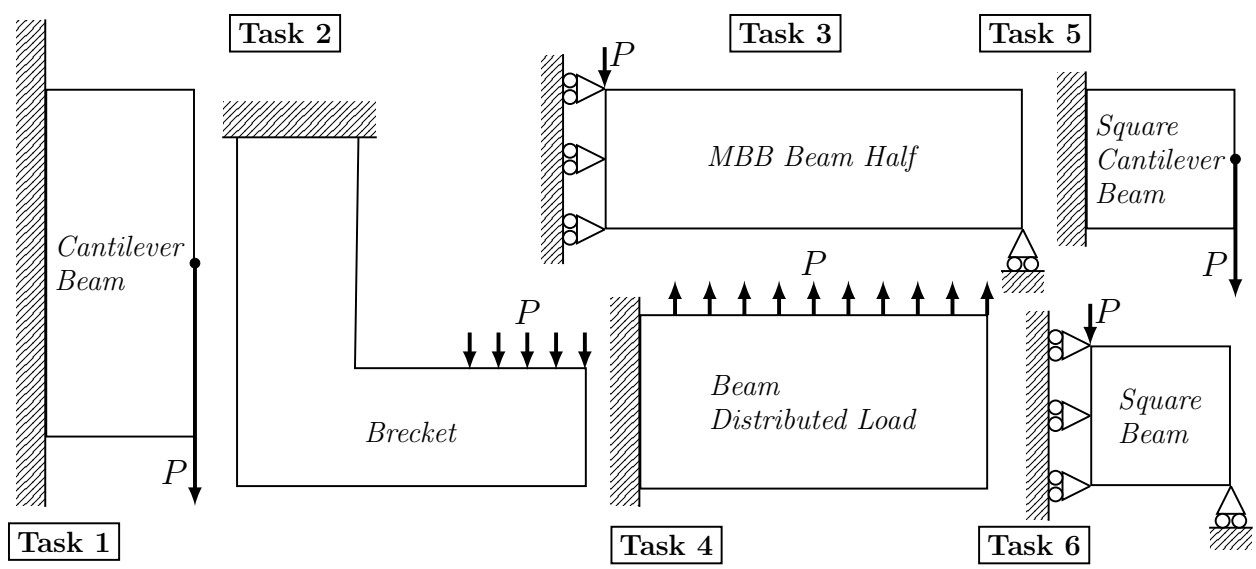

*Figure 11.* Schematics and solutions of the tasks.

$1\,\text{m} \times 1\,\text{m}$.

**Task 6: Square Beam**. This task replicates a previous task (Task 3), but the design domain has been modified to a square measuring $1\,\text{m} \times 1\,\text{m}$.

For all cases, the Poisson ratio and Young's modulus are set to $\nu = 0.3$ and $E = 1\,\text{Pa}$. Schematics of these cases are shown in Fig. 6, along with the solutions obtained using the SIMP method.

## E. Datasets

***Dataset for training the Heaviside Decoder*** consists of 5k samples for each type of geometric feature (ellipse, triangle, quadrilateral) with varying rounding radii. As illustrated in Fig. 3, each sample is designed to include the minimal number of variables for each feature. For ellipses, one axis is fixed at 0.5, and the axis ratio is used as the variable in $\chi$. For triangles and quadrilaterals, two vertices are fixed at coordinates $(-0.5, -0.25)$ and $(0.5, -0.25)$, respectively, while the coordinates of the remaining vertices are used in $\chi$ along with the rounding radii. For each geometric feature, $1\,000$ random points are generated within the square $[-1, 1] \times [-1, 1]$. The approximated Heaviside function is computed using Equation 8.

***Dataset for training the Reconstruction Decoder*** contains only 5 million samples of $\chi$ for each type of geometric feature.

***Dataset for testing the Heaviside Decoder*** is generated similarly to the training dataset, but it contains 500 samples for each type of geometric feature.

***Dataset for testing the Reconstruction Decoder*** is generated similarly to the training dataset, but it contains 10k samples for each type of geometric feature.

The ***surface test dataset*** contains 100 geometric features with 1225 points per feature, generated in a grid format. This dataset is used to compute the smoothness metric.

## F. T-test for Comparison of Different Model Architectures

Note:

- The test was conducted on metric evaluations after 20 model training runs. All seeds are not fixed.

- The p-value is shown in the graphs on the right, relative to the best method for each specific metric.

- "st1" means the first strategy, where the latent space is trained via the Heaviside decoder. "st2" means the second strategy, where the latent space is trained via the Reconstruction decoder.

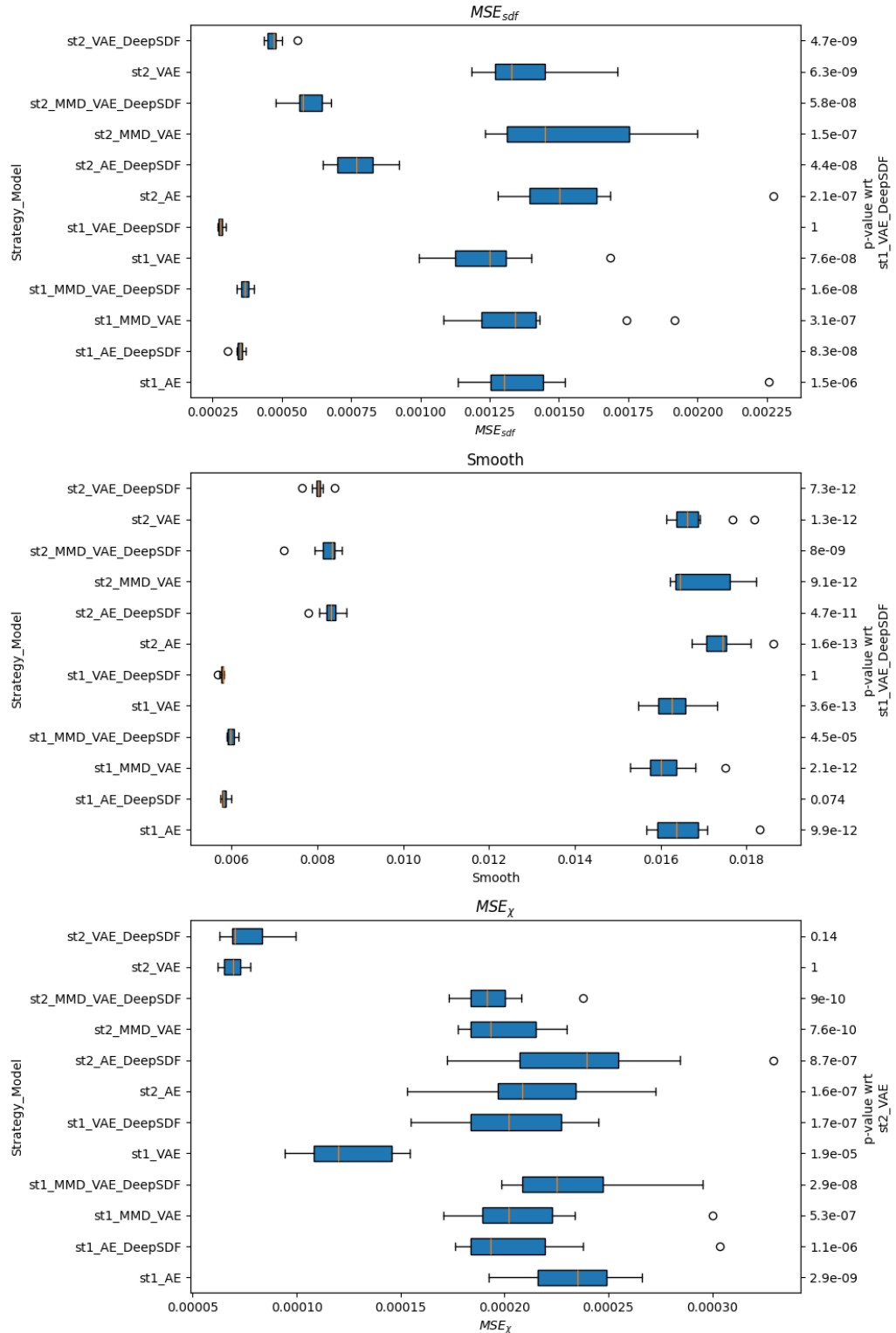

*Figure 12.* Boxplot for all training strategies and models.

## G. Kreisselmeier-Steinhauser (KS) Function, Implementation Details

It is important for us to maintain sensitivity with respect to the objective function; however, to ensure the method does not fail, we are forced to clamp the $\rho$ parameter because the KS function can produce values greater than 1. Therefore, in addition to equation (11), we use the following formula to scale $\rho$

$$\rho_e = \frac{1 - \rho_{\min}}{KS_{\max} - KS_{\min}} \left( KS_{\max} - \frac{\ln \sum_{m=1}^{M} \exp(\gamma_{KS} \widetilde{H_{m,e}})}{\gamma_{KS}} \right) + \rho_{\min} \tag{21}$$

where $KS_{\min}$ is the minimum value of the KS function, which is equal to

$$KS_{\min} = \frac{\ln P}{\gamma_{KS}} \tag{22}$$

and $KS_{\max}$ is the maximum value of the KS function, which is equal to

$$KS_{\max} = \frac{\ln(P \exp(\gamma_{KS}))}{\gamma_{KS}} \tag{23}$$

where $P$ is the expected maximum number of geometric primitives intersecting at one point (by default, this is 2).

Therefore, the combined $\rho$ will exceed the range $[\rho_{min}, 1]$ only at intersections where more than $P$ geometric primitives overlap. In these cases, $\rho$ is clamped to the range and becomes insensitive to the objective function.

In our case, the primary topology shape is a non-overlapping primitive in which a void must form. For large values of P but small values of $\gamma_{KS}$, the value of $\rho$ inside the primitive becomes considerably higher than $\rho_{min}$. This behavior is evident from the minimum $\rho$ values ($\min(\rho)$) reported in Table 5 for Task 3: MBB beam half, where no primitives intersect. Therefore, we choose $\gamma_{KS}$ to be as large as possible while ensuring that computations remain practical and that excessively large values of $\exp(\gamma_{KS})$ are avoided.

Changing $\gamma_{KS}$ above 10 has little impact on convergence or final topology, whereas lower values worsen convergence because $\rho_e$ remains too high for void formation.

We conducted experiments with different $\gamma_{ks}$ values for Example 3: Bracket. The results are shown in Table 6 and Figure 13.

*Table 5.* Experiments with different $\gamma_{ks}$ values for Task 3: MBB beam half

| $\gamma_{ks}$ | 10 | 20 | 30 | 40 | 50 | 60 | 70 | 80 |
|---|---|---|---|---|---|---|---|---|
| vf | 0.454 | 0.471 | 0.487 | 0.437 | 0.510 | 0.460 | 0.451 | 0.465 |
| C | 0.00182 | 0.00169 | 0.00151 | 0.00163 | **0.00141** | 0.00155 | 0.00156 | 0.00153 |
| $\min(\rho)$ | 0.06997 | 0.03533 | 0.02378 | 0.01800 | 0.01454 | 0.01223 | 0.01058 | 0.00934 |
| $\max(\rho)$ | 0.93014 | 0.96479 | 0.97634 | 0.98212 | 0.98559 | 0.98790 | 0.98955 | 0.99078 |

*Table 6.* Experiments with different $\gamma_{ks}$ values for Task 2: Bracket

| $\gamma_{ks}$ | 10 | 20 | 30 | 40 | 50 | 60 | 70 | 80 |
|---|---|---|---|---|---|---|---|---|
| vf | 0.373 | 0.358 | 0.347 | 0.355 | 0.347 | 0.344 | 0.357 | 0.354 |
| C | 0.00238 | 0.00212 | 0.00209 | 0.00207 | 0.00210 | 0.00213 | 0.00203 | **0.00198** |
| $\min(\rho)$ | 0.00067 | 0.00105 | 0.00504 | 0.00089 | 0.00116 | 0.00163 | 0.00623 | 0.00103 |
| $\max(\rho)$ | 0.87418 | 0.93682 | 0.95769 | 0.96813 | 0.97439 | 0.97857 | 0.98155 | 0.98379 |

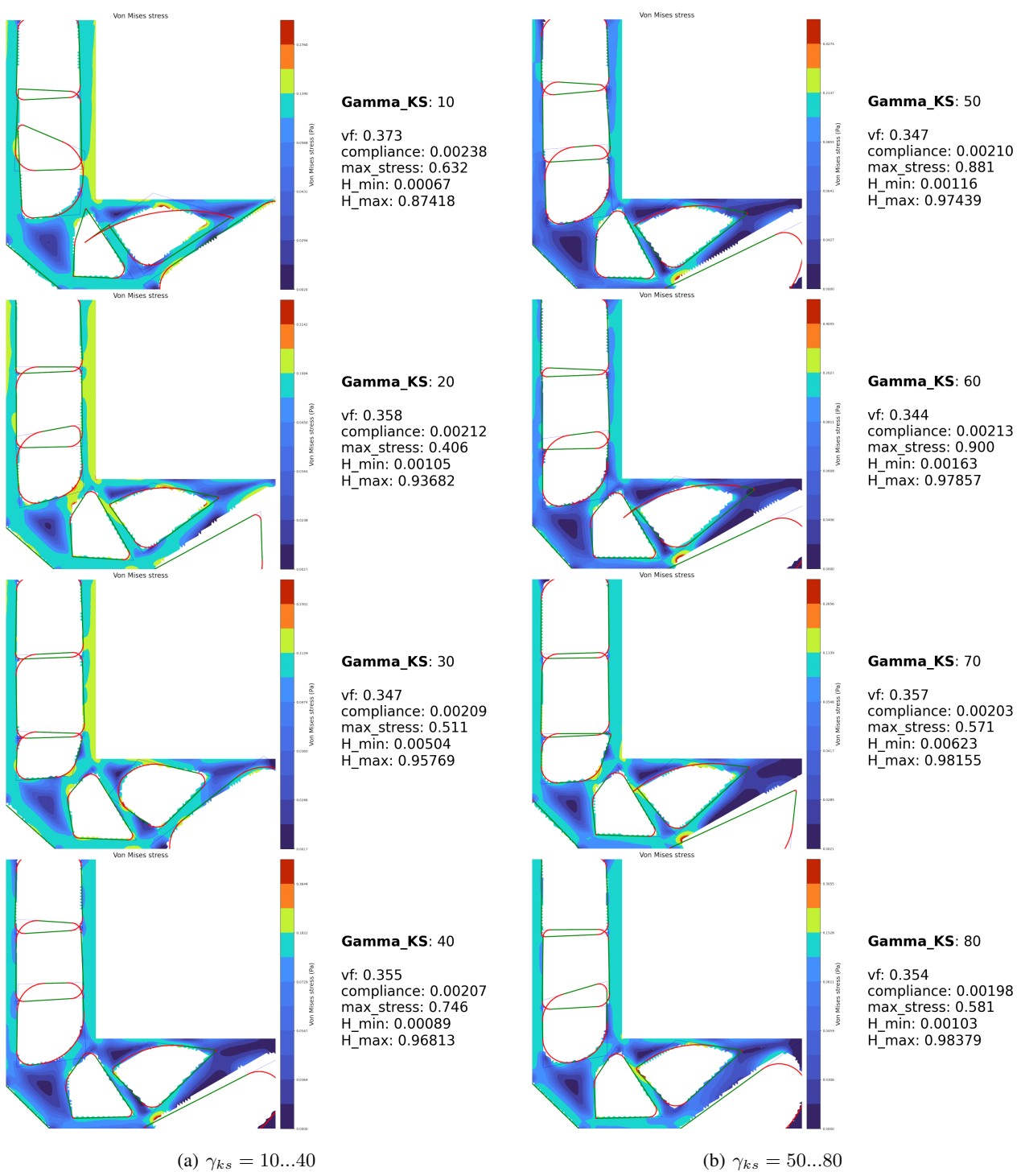

(a) $\gamma_{ks} = 10...40$

(b) $\gamma_{ks} = 50...80$

*Figure 13.* Brecket results for different $\gamma_{ks}$ ranges.

# H. Method Comparison

The task schemes are presented in Figure 11.

Notes:

- The presented von Mises plot is used here for a visual assessment of the quality of the internal stress distribution in the structure. Using its maximum values as a metric for comparing methods is impractical in this case.

- The specified $V_{max}$ value in FMTO is chosen to demonstrate that even with the smallest amount of material, our method achieves better compliance metric values.

- TreeTOP is implemented only for rectangular domains. Although the NTopo method allows for domain modifications, specifying the boundary conditions is complicated, and therefore we did not implement it. Consequently, the comparison for the "Bracket" problem includes only SIMP and Elliptical FMTO.

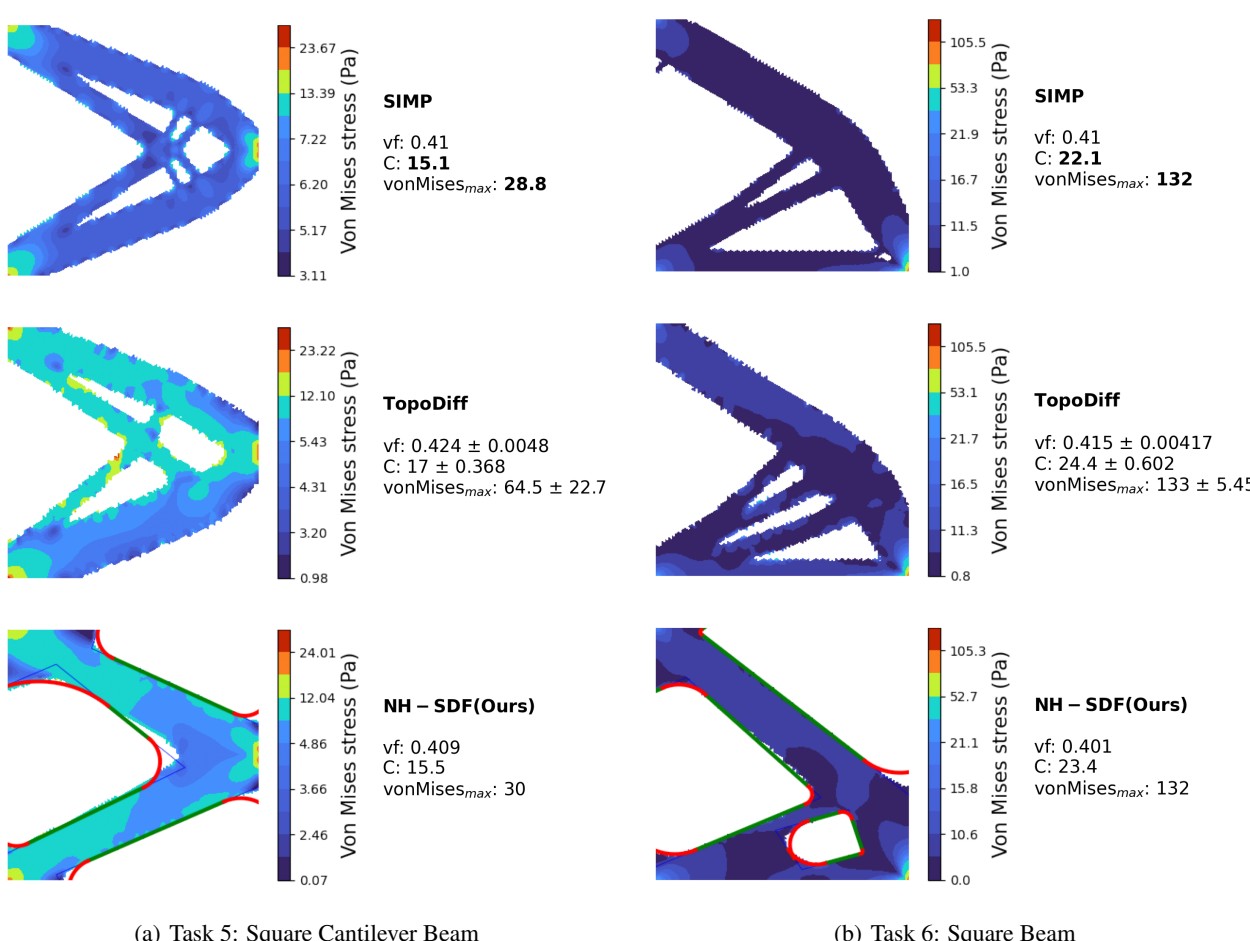

(a) Task 5: Square Cantilever Beam  (b) Task 6: Square Beam

*Figure 14.* Solutions obtained using SIMP, TopoDiff and Neural Heaviside SDF for Task 5 and Task 6.

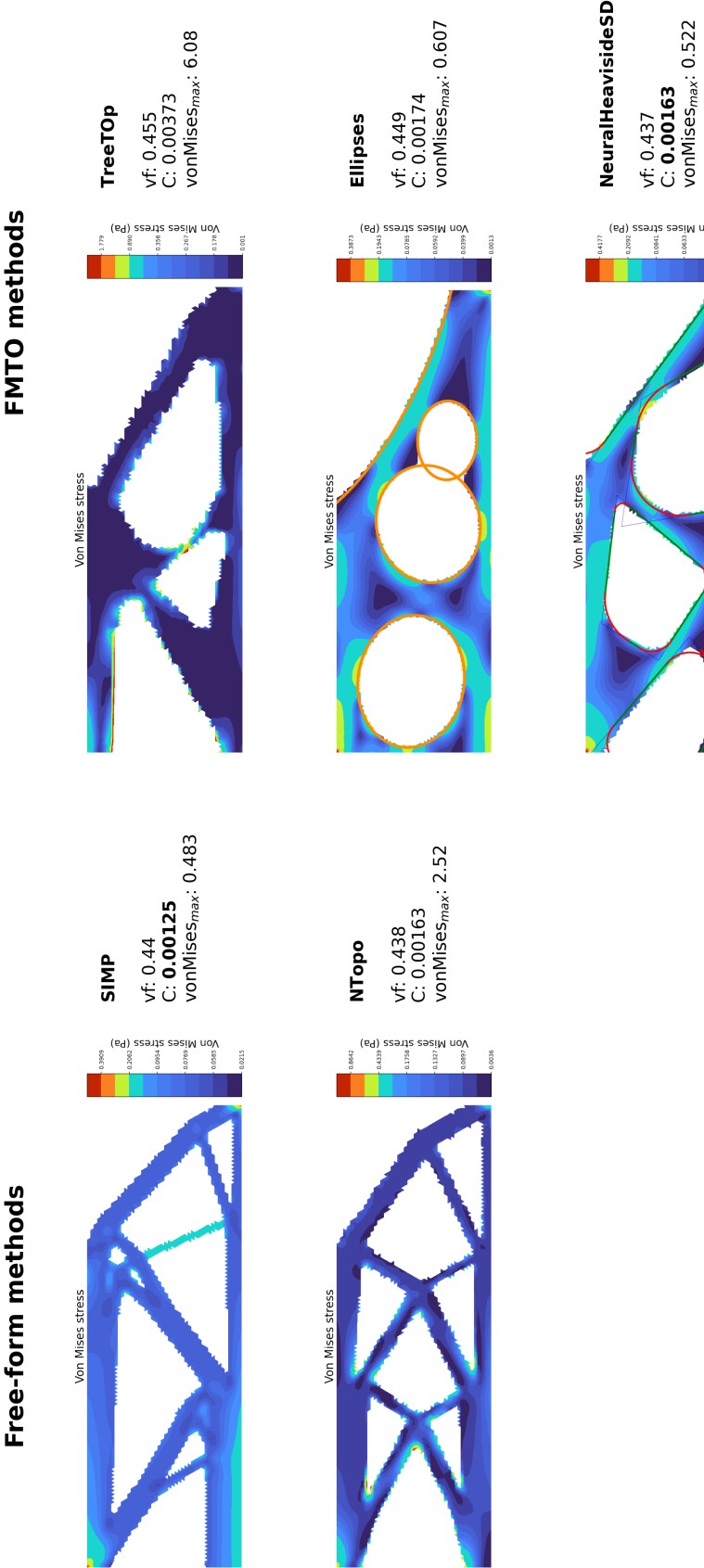

*Figure 15.* MBB Beam Half

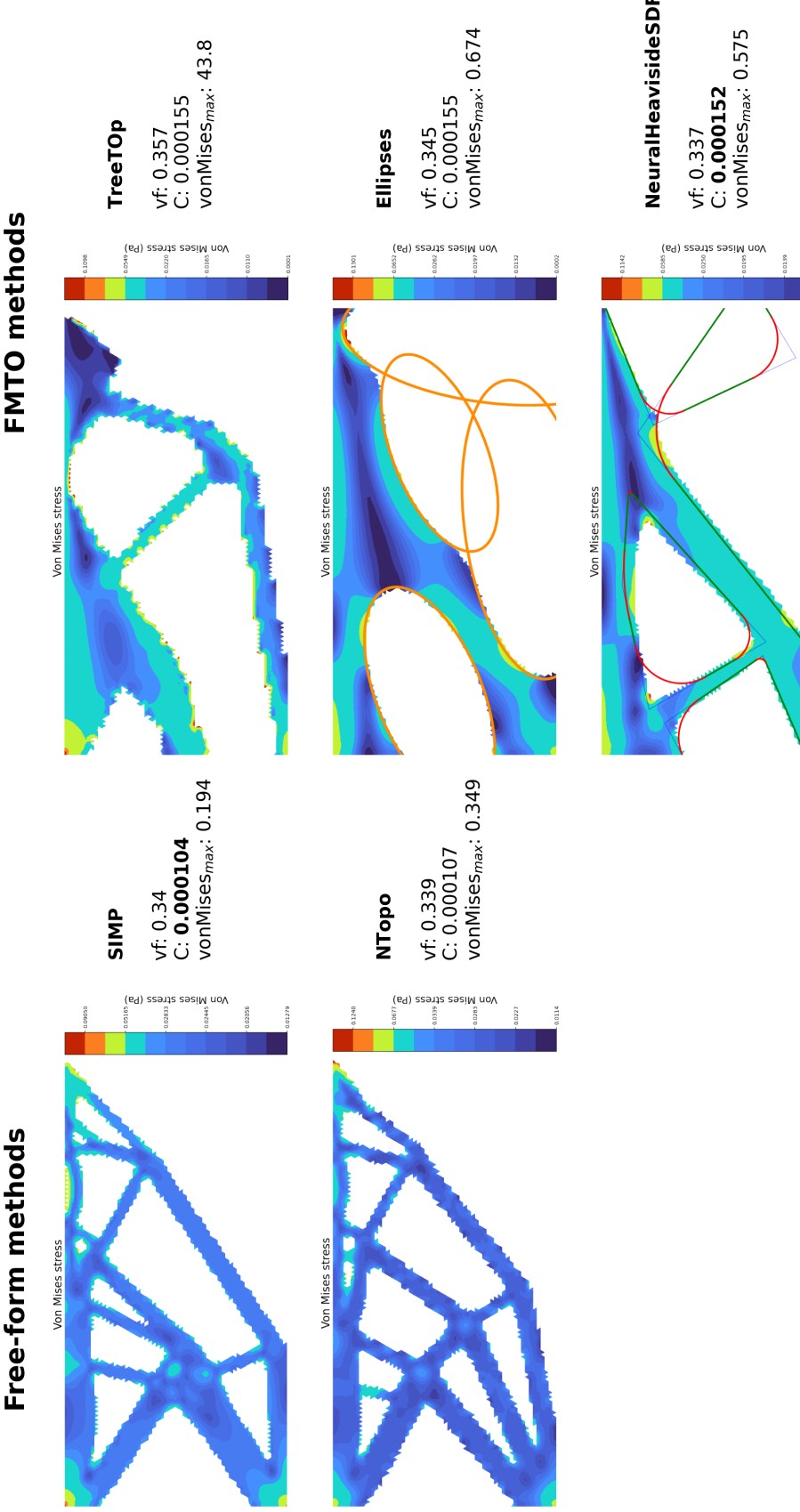

*Figure 16.* Beam Distributed Load

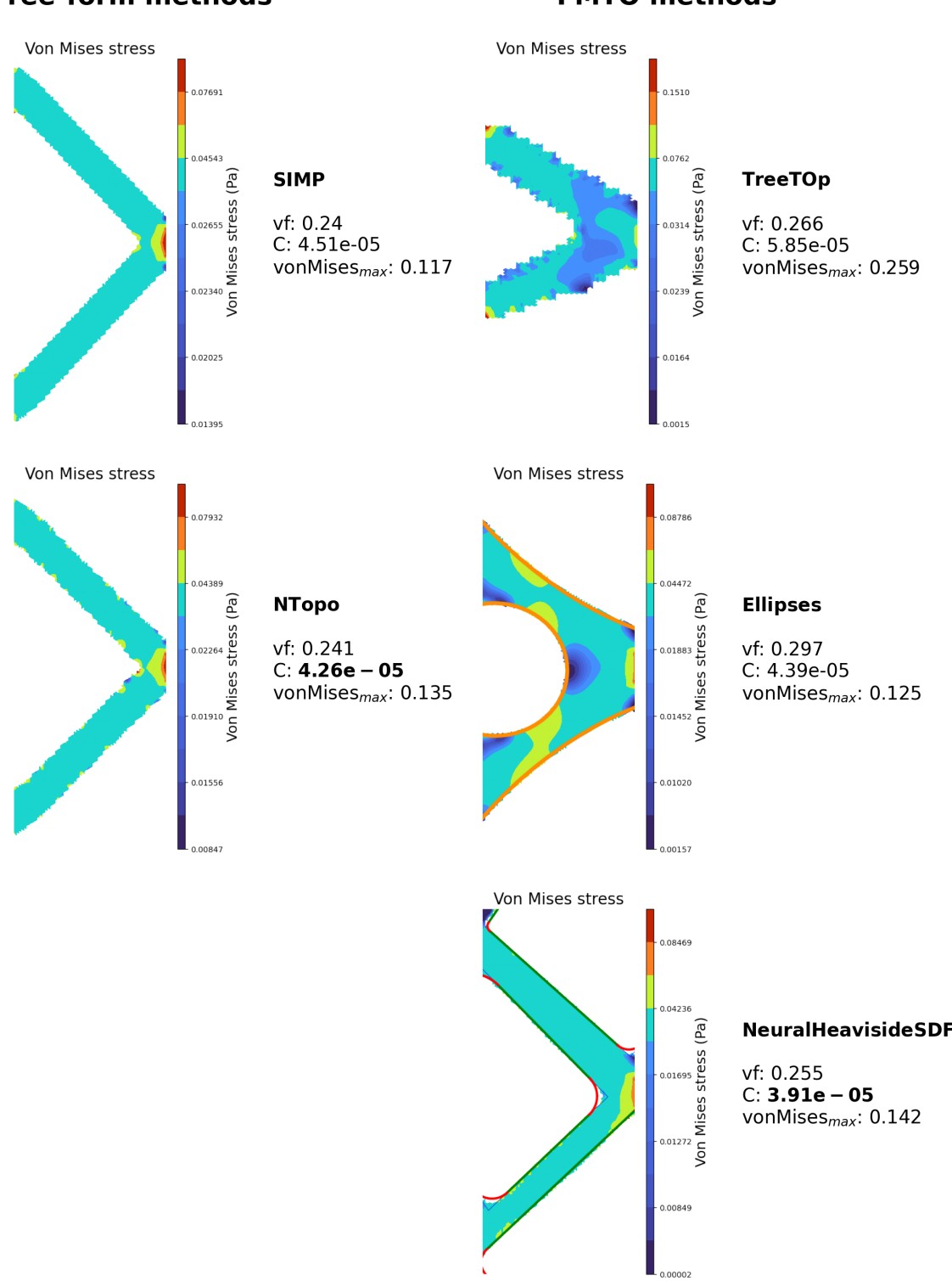

*Figure 17.* Cantilever Beam

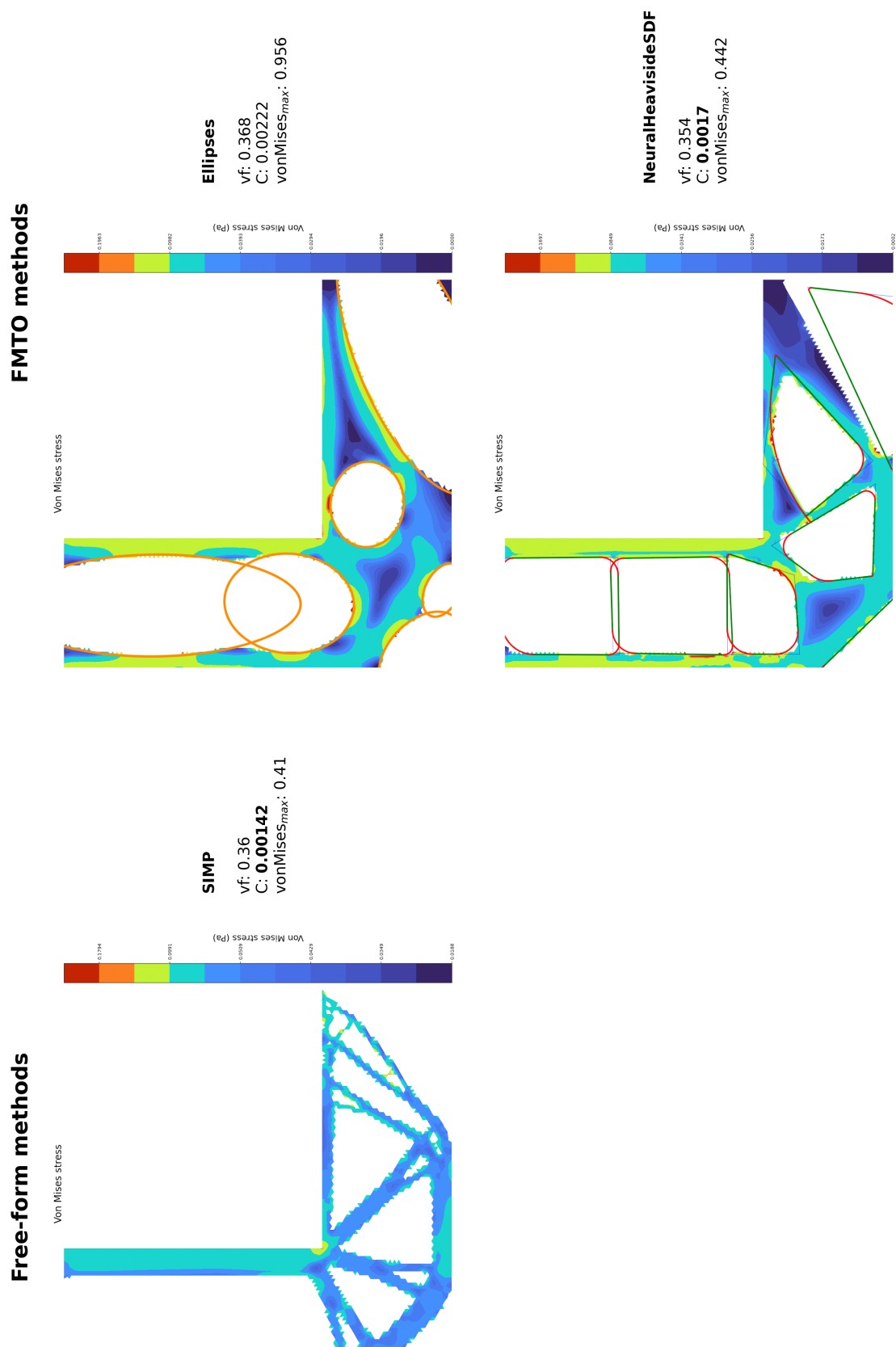

*Figure 18.* Bracket

