# OpenReview forum: "Feature-Mapping Topology Optimization with Neural Heaviside Signed Distance Functions"
_ICML.cc/2025/Conference — ICML 2025 poster_

### Official Review · Reviewer_ZcDR · 2025-03-09

**Overall Recommendation:** 4

**Summary:**

This paper presents a novel deep constrained topology optimization algorithm. Using the SIMP formalism, the authors propose learning an encoding of a space of fabricable shapes and solve the optimization problem on this space using gradient-based methods.

## update after rebuttal
In light of these comparisons and clarifications (and on the assumption that they all are included in the updated manuscript), I am raising my score. I hope this paper gets presented at ICML!

**Claims And Evidence:**

All claims made in the paper are thoroughly supported. It should be noted that these claims are measured: for example, the authors limit themselves to exploring several possible new frameworks for topology optimization, without necessarily claiming its superiority with respect to other works.

**Essential References Not Discussed:**

Like I said above, the authors are not expected to perform a full survey of this topic, but I would have appreciated including works from the engineering community (e.g., “Multiscale structural topology optimization with an approximate constitutive model for local material microstructure”) or the graphics one (e.g., “Two-Scale Topology Optimization with Microstructures”).

**Experimental Designs Or Analyses:**

See above

**Methods And Evaluation Criteria:**

The authors evaluate their method on a broad range of examples, which are described in the supplemental. They even include code in their supplemental material that reproduces most results.

In general, I was surprised that the authors did not evaluate and ablate their algorithmic choices more thoroughly. For example, this paper relies on encoding the shape code X into a (smaller) latent space Z, and then optimizes the code Z according to the physical problem using projected gradient descent, effectively ensuring that the optimal Z is the image through the encoding of a valid X. A question that may occur to a reader is: why is this encoding necessary? Can’t one simply do projected gradient descent on X, projecting (or “refactoring”, as said in the paper) X into the set of possible X after every 5 iterations? Given the encoding adds a lot of the complexity in the algorithm, I would have expected to have a clear answer to this question in the text or experiments, but I missed it.

Similarly, the paper uses a Heaviside-“SDF” representation for the primitives. This choice feels somewhat ad-hoc, justified by saying “[it] is convenient for the optimization process and for training the neural network”. I wished the authors either elaborated on this, or compared to other strategies like learning the non-Heaviside SDF directly (in fact, isn’t the Heaviside loss in Eq. 9 the same as an SDF loss with anisotropic sampling of B?).

**Other Comments Or Suggestions:**

“Cantilever” is misspelled in C.1, and the citation “Maz’e” should be “Mazé”.

**Other Strengths And Weaknesses:**

The main strength of the paper is its novelty: using learned latent spaces of shapes as a way of guiding topology optimization to fabricable shapes is a good idea, and this is a relatively good execution of it.

As said above, I believe the main weakness in this work is the lack of comparison to prior art (especially for a task as crowded as this), as well as the lack of experiments evaluating the algorithmic choices. These are the only reasons I am not more excited about accepting this work, although I will carefully read and review the authors’ response and other reviewer comments.

**Questions For Authors:**

- Can the authors please clarify the dependencies of each variable on rho in Sec 3.2.?
- What is \xi_e and how does it differ from \xi in Eqs. (7-8)
- Why are there 6 entries in \xi in Fig. 3? Isn’t \xi a (x,y,z) coordinate?
- The shape code X is never introduced, defined or explained beyond Figure 4 (which appears a full page after X is used in the text). Perhaps front loading the definition of X would be useful to a reader?

**Relation To Broader Scientific Literature:**

The paper does not (nor does it claim or intend to) include a full survey of work in topology optimization, a large field of research with decades of work. Instead, it focuses on deep feature-mapping methods for topology optimization and, as far as I can tell, does a good job of covering these.

Nonetheless, I am somewhat disappointed that the manuscript does not attempt to place this algorithm in context with other topology optimization strategies. Given that this is a problem considered in different fields of research and with many interested practicioners, I would have expected this paper to clearly answer the question “how is this algorithm better or worse than existing topology optimization methods?”, i.e., “if I were a manufacturing company, when if ever do I wish to use this method as opposed to others?” This can be answered in text or in experimentation.

**Theoretical Claims:**

See above

---

> ### Author Rebuttal · Authors · 2025-03-31
>
> Thank you for your thorough and insightful review! We are grateful for your detailed comments, which have not only highlighted the strengths of our methodology but also pointed out key areas for improvement, ultimately helping us enhance the clarity and impact of our research.
>
> ***Refactoring mechanism.***
> During topology optimization, reconstructed geometry can significantly deviate from the Heaviside function boundaries because the latent representation $ Z $ extends beyond the learned shape distributions. To address this, we initially reprojected latent vectors into the correct regions using an additional objective function, but this hindered convergence. Our refactoring mechanism instead steers latent vectors back to their learned distribution and potentially allows for geometric corrections during optimization, such as enforcing minimum radii or correcting overlapping arcs.
>
> Of course, we did not try every possible variant that might address the shortcomings of our method. In the future, we plan to explore alternative approaches to improve the algorithm.
>
> ***Heaviside-“SDF” representation.*** We chose to use the Heaviside-SDF representation instead of a direct SDF during model training for the following reasons:
>
> 1. The Heaviside transformation produces a sharper, more distinct boundary—exactly what we need for our application.
>
> 2. This transformation enhances FMTO's robustness to neural network noise. Using a direct SDF requires applying the Heaviside function later, which significantly amplifies boundary noise occurring in the model.
>
> ***Motivation for the method.***
> Our method is motivated by the common industry practice of creating parts by extruding sketches—typically polygons with rounded corners—for milling and casting. We developed a framework that allows polygons to evolve from an initial circle during optimization, producing more natural and manufacturable designs. Additionally, we plan to extend the framework to include other common geometric features, such as polygons with arc segments.
>
> ***Essential References.*** Thank you for the note. These directions are indeed closely related to our work, and we would be happy to include them in our review.
>
> ***Comparison with other methods.*** We can provide more comparisons with other methods, including  one recently published work $\texttt{TreeTOp}$ [1], which directly relates to FMTO.
>
> 1. K. Padhy, R., Thombre, P., Suresh, K. et al. Treetop: topology optimization using constructive solid geometry trees. 2025.
>
> | Method | Method type | $\text{vonMises}_{max}$ |  Compliance | Volume Fraction |
> | :--- | :---: | :---: | :---: | :---: |
> | SIMP | Free-form | $\textbf{0.483}$ | $\textbf{0.00125}$ | 0.44 |
> | NTopo | Free-form | 2.52 | 0.00163 | 0.438 |
> | TreeTOp | FMTO | 6.08 | 0.00373 | 0.455 |
> | Ellipses | FMTO | 0.607 | 0.00174 | 0.449 |
> | NeuralHeavisideSDF | FMTO | $\textbf{0.522}$ | $\textbf{0.00163}$ | 0.437 |
>
> Table: Methods comparison for Example 3: MBB beam half
>
> | Method | Method type | $\text{vonMises}_{max}$ |  Compliance | Volume Fraction |
> | :--- | :---: | :---: | :---: | :---: |
> | SIMP | Free-form | $\textbf{0.194}$ | $\textbf{0.000104}$ | 0.34 |
> | NTopo | Free-form | 0.349 | 0.000107 | 0.339 |
> | TreeTOp | FMTO | 43.8 | 0.000155 | 0.357 |
> | Ellipses | FMTO | 0.674 | 0.000155 | 0.345 |
> | NeuralHeavisideSDF | FMTO | $\textbf{0.575}$ | $\textbf{0.000152}$ | 0.337 |
>
> Table: Methods comparison for Example 4: Beam Distributed Load
>
> $\textbf{Note:}$ Since the NTopo method is quite limited in altering the parameters of the initial conditions, we had to adjust the parameters of the other methods to match those of the NTopo method. Therefore, the metric results differ from those in the original manuscript.
>
> Our approach achieves the best Compliance metric values among all FMTO methods while using less material. Additionally, in some experiments, our method is comparable to the NTopo method (see Example 3).
>
> ***“Cantilever” is misspelled, incorrect citation, “linear elasticity” keyword.*** Thank you for pointing this out. We will correct the spelling error, add the “linear elasticity” keyword, and fix the citation.
>
> ***What is $\xi_e$ and how does it differ from $\xi$ in Eqs. (7-8).*** $\xi$ represents any point within the design domain and is used to train the model. In contrast, $\xi_e$ denotes the center point of element $e$.
>
> ***Why are there 6 entries in $\xi$ in Fig. 3? Isn’t $\xi$ a (x,y,z) coordinate?*** We apologize for the confusion caused by our previous depiction of a batch size of 3. We have updated the scheme to display a batch size of 1. Our experiments focus on 2D problems, so only two coordinates (x, y) are used.
>
> ***Shape code $X$.*** We apologize for the missing definitions. Inconsistencies in the method description have been corrected, and an updated version—including details on the shape code $\chi$—will appear in the main text, as well as a revised Figure 4. Additionally, a table of notations will be added to the appendix.

---

> > ### Comment · Reviewer_ZcDR · 2025-04-04
> >
> > Thank you very much! In light of these comparisons and clarifications (and on the assumption that they all are included in the updated manuscript), I am raising my score. I hope this paper gets presented at ICML!

---

> > > ### Author Response · Authors · 2025-04-04
> > >
> > > Thank you very much for your thorough evaluation and for reconsidering your score. We deeply appreciate the valuable insights you have provided. We assure you that the suggested changes will be incorporated into the final version of both the paper and the accompanying code.
> > >
> > > Thank you again for your support and constructive feedback!

---

### Official Review · Reviewer_UvMW · 2025-03-10

**Overall Recommendation:** 3

**Summary:**

This paper designs a new learning-based approach for feature mapping based topology optimization. The major advantage of the proposed method against previous works is that the generated voids are guaranteed to be directly manufacturable, thus circumventing cumbersome post-processing procedures. Technically, the integration of Neural Heaviside SDFs with structured latent spaces addresses the limitation of traditional feature-mapping methods by enabling diverse geometric features, thus improving manufacturability. In general, this characteristic well aligns with actual industrial needs.

**Claims And Evidence:**

Partially.

**Essential References Not Discussed:**

No.

**Experimental Designs Or Analyses:**

Yes.

**Methods And Evaluation Criteria:**

Yes.

**Other Comments Or Suggestions:**

N/A

**Other Strengths And Weaknesses:**

As presented in Eq. (8), the Sigmoid function tuned by \beta is used as a soft approximation of the Heaviside function. I think throughout the paper it would be more straightforward to directly use the description of "Sigmoid", instead of "Heaviside".

Empirically, the authors experimented with examples of ellipses. triangles, and quadrilaterals. I am wondering if more types of geometric primitives with higher degrees of complexities can be included for evaluation.

Eq. (12) introduces Kreisselmeier-Steinhauser (KS) function for smooth max approximation, but sensitivity to the smoothing parameter γ_KS is not analyzed. How does γ_KS affect convergence?

The ellipse-based baseline is limited. Why not compare against B-spline or Bézier-based feature-mapping methods?

**Questions For Authors:**

N/A

**Relation To Broader Scientific Literature:**

Well related to prior studies on topology optimization and geometry processing.

**Theoretical Claims:**

Yes.

---

> ### Author Rebuttal · Authors · 2025-03-31
>
> Thank you very much for taking the time to provide such a detailed and insightful review. Your comment has been extremely valuable in enabling us to carefully analyze various aspects of our method's implementation, including the integration of the Kreisselmeier-Steinhauser (KS) function.
>
> ***Using Sigmoid instead of Heaviside.*** The use of the term "sigmoid" is, of course, acceptable. However, the Heaviside function is a more traditional term in the context of topology optimization. We, like many other authors in this field, prefer to retain the possibility of using various smooth approximations of the Heaviside function without changing the name of the method.
>
> ***Higher Degrees of Complexity.*** Adding new primitives forces an increase in latent space dimensions to keep accuracy, which can harm convergence. However, since our latent space captures overall contours rather than exact parameters, we believe that there's a threshold beyond which more dimensions are unnecessary. We will investigate this further.
>
> ***Kreisselmeier-Steinhauser (KS) function.*** The current implementation includes several additional details that are not covered in the main text.
>
> It is important for us to maintain sensitivity with respect to the objective function; however, to ensure the method does not fail, we are forced to clamp the $\rho$ parameter because the KS function can produce values greater than 1. Therefore, in addition to equation (11), we use the following formula to scale $\rho$:
>
> $$
> \\rho_e = \\frac{1 - \\rho_{\\min}}{KS_{\\max} - KS_{\\min}} \\left(KS_{\\max} - \\frac{\\ln \\sum_{m=1}^{M} \\exp(\\gamma_{KS} \\widetilde{H_{m,e}})}{\\gamma_{KS}}\\right) + \\rho_{\\min}
> $$
>
> where $ KS_{\min} $ is the minimum value of the KS function, which is equal to
>
> $$
>     KS_{\min} = \\frac{\\ln P}{\\gamma_{KS}}
> $$
>
> and $ KS_{\max} $ is the maximum value of the KS function, which is equal to
>
> $$
>     KS_{\\max} = \\frac{\\ln (P\\exp(\\gamma_{KS}))}{\\gamma_{KS}}
> $$
>
> where $P$ is the expected maximum number of geometric primitives intersecting at one point (by default, this is 2).
>
> Therefore, the combined $\rho$ will exceed the range $[\rho_{min}, 1]$ only at intersections where more than $P$ geometric primitives overlap. In these cases, $\rho$ is clamped to the range and becomes insensitive to the objective function.
>
> In our case, the primary shape for topology is a non-overlapping primitive within which a void must form. For large values of P but small values of $\gamma_{KS}$, the value of $\rho$ inside the primitive becomes considerably higher than $\rho_{min}$. Therefore, we choose $\gamma_{KS}$ to be as large as possible, while ensuring that computations do not become impractical due to excessively large values of $\exp(\gamma_{KS})$.
>
> Changing $\gamma_{KS}$ above 10 has little impact on convergence or final topology, whereas lower values worsen convergence because $\rho_e$ remains too high for void formation.
>
> $\gamma_{ks}$ | 10 | 20 | 30 | 40 | 50 | 60 | 70 | 80
> ----- | ----- | ----- | ----- | ----- | ----- | ----- | ----- | -----
> vf | 0.373 | 0.358 | 0.347 | 0.355 | 0.347 | 0.344 | 0.357 | 0.354
> C | 0.00238 | 0.00212 | 0.00209 | 0.00207 | 0.00210 | 0.00213 | 0.00203 | **0.00198**
> $min(\rho)$ | 0.00067 | 0.00105 | 0.00504 | 0.00089 | 0.00116 | 0.00163 | 0.00623 | 0.00103
> $max(\rho)$ | 0.87418 | 0.93682 | 0.95769 | 0.96813 | 0.97439 | 0.97857 | 0.98155 | 0.98379
>
> Table: experiments with different $\gamma_{ks}$ values for Example 3: Bracket
>
> ***B-spline and Bézier-based Feature-Mapping Methods.*** Unfortunately, most published works in topology optimization come without accompanying code and do not always provide enough details for reproduction. However, we have the opportunity to compare our method with a recently published approach [1], where the main geometric feature is a polygon constructed from half-spaces. These comparisons will be added to the manuscript.
>
> [1] K. Padhy, R., Thombre, P., Suresh, K. et al. Treetop: topology optimization using constructive solid geometry trees.
>
> | Method | Method type | $\text{vonMises}_{max}$ |  Compliance | Volume Fraction |
> | :--- | :---: | :---: | :---: | :---: |
> | SIMP | Free-form | $\textbf{0.483}$ | $\textbf{0.00125}$ | 0.44 |
> | NTopo | Free-form | 2.52 | 0.00163 | 0.438 |
> | TreeTOp | FMTO | 6.08 | 0.00373 | 0.455 |
> | Ellipses | FMTO | 0.607 | 0.00174 | 0.449 |
> | NeuralHeavisideSDF | FMTO | $\textbf{0.522}$ | $\textbf{0.00163}$ | 0.437 |
>
> Table: Methods comparison for Example 3: MBB beam half
>
> $\textbf{Note: }$ Since the NTopo method is quite limited in altering the parameters of the initial conditions, we had to adjust the parameters of the other methods to match those of the NTopo method. Therefore, the metric results differ from those in the original manuscript.
>
> Our approach achieves the best Compliance metric values among all FMTO methods while using less material. Additionally, in some experiments, our method is comparable to the NTopo method (see Table with Example 3).

---

> > ### Comment · Reviewer_UvMW · 2025-04-08
> >
> > Thanks for the authors' responses and explanations. Some of my concerns and questions are explicitly addressed. However, due to my inadequare expertise in this field, I am afraid that I cannot give further higher score for this paper.

---

### Official Review · Reviewer_ZNau · 2025-03-12

**Overall Recommendation:** 2

**Summary:**

The authors propose a novel neural approximation framework based on a variational autoencoder (VAE) model to approximate the Heaviside function of the Signed Distance Function (SDF), enabling a unified representation of diverse geometric features in a single latent space. This approach integrates machine learning with traditional topology optimization (TO) to overcome the limitations of existing TO methods. The key contribution includes an improved optimization framework that incorporates volume constraints into the objective function like the Kreisselmeier-Steinhauser (KS) function for smooth maximum function approximation, and efficient gradient computation and sensitivity analysis using adjoint differentiation.

**Claims And Evidence:**

Yes.

**Essential References Not Discussed:**

Yes. Please refer to the weaknesses part.

**Experimental Designs Or Analyses:**

Yes.

**Methods And Evaluation Criteria:**

Yes.

**Other Comments Or Suggestions:**

Please see the above weaknesses part.

**Other Strengths And Weaknesses:**

**Strengths**:

- The authors conducted several experiments using a set of reasonable metrics to evaluate the accuracy of predictions (MSE) and measure the noise of the gradient on grid points (Smoothness Metrics).

- The authors implemented an approach similar to an ellipse-based method that achieves superior compliance values. While the SIMP method creates small, locally conditioned edges through a free-form approach, the proposed method avoids and ensures manufacturability.

**Weaknesses**
- VAE encode-decode structure: The use of VAE to learn geometric feature distributions has been extensively explored [1, 2], even in 3D scenes [3]. Learning the shape distribution is a commonly used approach that does not introduce efficient improvements or novel insights.

- Neural Heaviside SDF: SDF, Heaviside, and sigmoid representations are the most basic techniques. Although they express geometric boundaries and allow for training of discrete spatial variables with continuous representations, similar functions like sigmoid and Heavide also widely improved and not taken as sufficient innovations.

- Noise in training and geometric approximation: Does the latent space representation derived from the VAE struggle to capture smooth transitions between certain geometric features (e.g., ellipses to polygons)? And considering the SDF precision, does it perform well in resolving sharp edges or high-curvature regions? How to avoid or decrease inaccuracies in boundary fitting.

[1] Chadebec, C., & Allassonnière, S. (2022). A geometric perspective on variational autoencoders. Advances in Neural Information Processing Systems, 35, 19618-19630.

[2] Vadgama, S., Tomczak, J. M., & Bekkers, E. J. (2022, November). Kendall shape-VAE: Learning shapes in a generative framework. In NeurIPS 2022 Workshop on Symmetry and Geometry in Neural Representations.

[3] Kosiorek, A. R., Strathmann, H., Zoran, D., Moreno, P., Schneider, R., Mokrá, S., & Rezende, D. J. (2021, July). Nerf-vae: A geometry aware 3d scene generative model. In International conference on machine learning (pp. 5742-5752). PMLR.

**Questions For Authors:**

Please see the above weaknesses part.

**Relation To Broader Scientific Literature:**

In my opinion, this work is more suitable for industrial or material journals or conferences.

**Theoretical Claims:**

There is no theoretical claim.

---

> ### Author Rebuttal · Authors · 2025-03-31
>
> Thank you for your comprehensive and constructive feedback on our work. We value your insights on the VAE encode-decode structure, the Neural Heaviside SDF representation, and the potential challenges related to geometric approximation and boundary precision.
>
> **Novelty of the Proposed Method.** We acknowledge that VAEs have been widely used for learning shape distributions. However, our work differentiates itself by integrating a learned neural approximation of the Heaviside SDF function within Feature Mapping Topology Optimization (FMTO) tasks. Conventional FMTO approaches rely on explicitly defined SDFs, restricting the geometric diversity that can be effectively represented. Our approach, in contrast, enables a unified latent space where multiple geometric primitives can coexist and transform smoothly, facilitating shape evolution in FMTO beyond conventional methods.
>
> Moreover, although sigmoid and Heaviside functions are commonly used for defining geometric boundaries, our method employs a learned neural surrogate. This allows for an adaptive, data-driven boundary representation, enhancing both optimization flexibility and saving manufacturability.
>
> **Smooth Transitions Between Geometric Features.** Your concern regarding smooth transitions, particularly between ellipses and polygons, is well-taken.
>
> While our dataset does not explicitly include gradual shape interpolations between such forms, our model demonstrates an ability to approximate smooth deformations even beyond the trained shape classes. As illustrated in Figure 6 of the main text, the learned latent space permits meaningful shape variations, even if exact interpolations between distinct classes are not explicitly encoded. Nevertheless, we recognize that the latent space could be further refined to better support smooth morphing between different geometric classes. Future work may explore improved training strategies, such as incorporating intermediate transition shapes or using additional regularization to better structure the latent space for smooth interpolations.
>
> **SDF Precision and Boundary Fitting.** We understand the importance of accurately capturing high-curvature regions and sharp edges in SDF representations. To reduce inaccuracies in boundary fitting, we have enhanced our dataset generation strategy as follows:
>
> Each shape type in our dataset now includes 5,000 instances, sampled to ensure comprehensive coverage of geometric variations.
>
> Additionally, we have modified the point generation method. Out of 10,000 points per shape, one third are now generated directly along the shape boundaries, improving precision and ensuring that the trained SDF captures intricate features more effectively. The remaining points are evenly distributed between those drawn from a Gaussian distribution centered around the shape and those concentrated near vertices and edges.
>
> These improvements have significantly enhanced boundary fidelity, minimizing discrepancies between original and reconstructed shapes. The updated dataset and training approach result in more accurate SDF modeling, especially in areas with high curvature and sharp transitions.
>
> **Discussion on Related Work.** We appreciate the reviewer’s suggestion to better position our contributions within the existing literature. While previous works [1,2,3] have showcased the use of VAEs for shape learning, our study uniquely incorporates this framework into FMTO with a learned Heaviside SDF representation. We will expand our discussion in the manuscript to clearly contrast our method with prior approaches and emphasize our contributions more explicitly.
>
> Once again, we sincerely thank the reviewer for their valuable feedback. We believe that our revisions effectively address the concerns raised and further strengthen our work.

---

> > ### Comment · Reviewer_ZNau · 2025-04-05
> >
> > Thanks for the authors' response. After carefully reading the authors' response and other reviewers' comments, I believe my judgement that the limited and incremental contribution of this work, particularly that similar methods or ideas have been extensively explored. Concerning the high level standard of ICML, I hold on my rating.

---

### Official Review · Reviewer_rbNQ · 2025-03-24

**Overall Recommendation:** 3

**Summary:**

In this work, the authors work on topology optimization, specifically, they propose a deep learning method to simulate Feature-Mapping Topology Optimization (FMTO) (and not SIMP). They propose two decoders, one for the reconstruction and another to approximate the heaviside function. They show results on variations of autoencoder architecture types and training strategies.

## update after rebuttal
After the rebuttal, I am increasing my score. From the experiments, the proposed work brings significant improvements compared to existing work such as free-form methods NTopo and TopoDiff, and other FMTO methods. Using standard shapes eases manufacturing, and in the rebuttal for other reviewers, the authors discuss how they can improve boundary fitting further.

**Claims And Evidence:**

See Experiments section

**Essential References Not Discussed:**

As mentioned above, the authors need to compare with existing deep learning based methods. The authors could also consider comparing with the following:
- Zhang, Zeyu, et al. "Topology optimization via implicit neural representations." Computer Methods in Applied Mechanics and Engineering 411 (2023): 116052.
- Zelickman, Yakov, and James K. Guest. "Introducing a general polygonal primitive for feature mapping-based topology optimization."
- Chi, Heng, et al. "Universal machine learning for topology optimization." Computer Methods in Applied Mechanics and Engineering 375 (2021): 112739.
- Sosnovik, Ivan, and Ivan Oseledets. "Neural networks for topology optimization." Russian Journal of Numerical Analysis and Mathematical Modelling 34.4 (2019): 215-223.

**Experimental Designs Or Analyses:**

- In this work, the authors propose using a Heaviside Decoder for the topology-optimization task. They have two decoders, different options for each decoder (DeepSDF or Symmetric) and different training strategies (train reconstruction before heaviside or vice-versa). The experiments compare their method with SIMP solver and FMTO (Ellipse) which are non-deep learning methods. While the proposed method doesn’t reach the levels of SIMP, it performs better than FMTO.
- The authors did extensive ablation study on different decoder types and strategies (Table 1) and compared against SIMP and FMTO in Table 2. However, the authors do not compare with any other deep learning based method. Specifically, in L31 and L36 they discuss diffusion-based topology-optimization methods by Maze & Ahmed’22 and Mohseni & Khodaygan’24. The authors do not provide any discussion of why existing methods cannot be compared to theirs. Hence, such baselines should be included.
- The model is trained as a reconstruction task, and the authors provide the training objective functions. However, how are the constraints like $V_{max}$, $s_{min}$, $s_{max}$ enforced during inference? They seem to be used only during the training objective function to penalize the outputs, and it is unclear how they are incorporated during inference/test.
- In Table 1, authors should bold the best value under each metric. Currently, they omit bolding when their proposed method doesn’t achieve the best performance.
- In Table 1 and Table 2, the authors should perform t-test to determine if the performance improvement is statistically significant or not.
- I appreciate the code release in the supplementary

**Methods And Evaluation Criteria:**

See Experiments section

**Other Comments Or Suggestions:**

See above.

**Other Strengths And Weaknesses:**

The presentation of the paper needs a lot of work. The authors tend to use notations without defining them, which makes the paper very difficult to read; the reviewer has to guess themselves what the notation means. I list a few below, but there were several such issues:
- Equation 7 uses $\Omega$ but never defines it. It is likely the boundary of the feature.
- What is H in the left side of Figure 3? It is not defined in text. Only $\tilde{H}$ and $\tilde{H}_{true}$ are defined in the text, but not H.
- No clear separation between the training and inference procedure.
- $X$ is not explicitly defined in the text. It is first used in L208, and the reader has to go to Fig. 4 to understand what it represents.

**Questions For Authors:**

I would re-consider my score after the authors' responses to my comments, and after discussion with other reviewers.

**Relation To Broader Scientific Literature:**

The work is important because most CAD-integrated design requires human intervention in the post-processing stage. This work aims to minimize human intervention.

**Theoretical Claims:**

N/A

---

> ### Author Rebuttal · Authors · 2025-03-31
>
> Thank you for your thorough and insightful review!! We deeply appreciate the time and effort you invested in evaluating our work.
>
> ***Comparison with other methods*** Our method follows the FMTO approach by creating topology with geometric primitives. Unlike traditional free-form optimization that focuses only on compliance, FMTO constrains voids to specific shapes. Thus, comparing with other FMTO methods is most appropriate, even though many lack available code and full reproducibility details.
>
> Nevertheless, we can still provide more comparisons with four methods, including two new frameworks: the first competitor is a free-form method based on deep learning, $\texttt{NTopo}$ [1], and one  recently published work $\texttt{TreeTOp}$ [2], which directly relates to FMTO. These comparisons will be added to the manuscript.
>
> | Method | Method type | $\text{vonMises}_{max}$ |  Compliance | Volume Fraction |
> | :--- | :---: | :---: | :---: | :---: |
> | SIMP | Free-form | $\textbf{0.483}$ | $\textbf{0.00125}$ | 0.44 |
> | NTopo | Free-form | 2.52 | 0.00163 | 0.438 |
> | TreeTOp | FMTO | 6.08 | 0.00373 | 0.455 |
> | Ellipses | FMTO | 0.607 | 0.00174 | 0.449 |
> | NeuralHeavisideSDF | FMTO | $\textbf{0.522}$ | $\textbf{0.00163}$ | 0.437 |
>
> Table: Methods comparison for Example 3: MBB beam half
>
> | Method | Method type | $\text{vonMises}_{max}$ |  Compliance | Volume Fraction |
> | :--- | :---: | :---: | :---: | :---: |
> | SIMP | Free-form | $\textbf{0.194}$ | $\textbf{0.000104}$ | 0.34 |
> | NTopo | Free-form | 0.349 | 0.000107 | 0.339 |
> | TreeTOp | FMTO | 43.8 | 0.000155 | 0.357 |
> | Ellipses | FMTO | 0.674 | 0.000155 | 0.345 |
> | NeuralHeavisideSDF | FMTO | $\textbf{0.575}$ | $\textbf{0.000152}$ | 0.337 |
>
> Table: Methods comparison for Example 4: Beam Distributed Load
>
> $\textbf{Note:}$ Since the NTopo method is quite limited in altering the parameters of the initial conditions, we had to adjust the parameters of the other methods to match those of the NTopo method. Therefore, the metric results differ from those in the original manuscript.
>
> Our approach achieves the best Compliance metric values among all FMTO methods while using less material. Additionally, in some experiments, our method is comparable to the NTopo method (see Example 3).
>
> 1. J. Zehnder, Y. Li, S. Coros, and B. Thomaszewski. NTopo: mesh-free topology optimization using implicit neural representations. 2021.
>
> 2. K. Padhy, R., Thombre, P., Suresh, K. et al. Treetop: topology optimization using constructive solid geometry trees. 2025.
>
> ***Training and inference*** During training, we optimize our model to best approximate the Heaviside SDF for various geometric primitives, independent of topology optimization. At inference, we use the frozen Heaviside decoder for SDF evaluation in FMTO and for geometry reconstruction during post‐processing.
>
> Constraints $V_{max}$, $s_{max}$, and $s_{min}$ are enforced at inference—$V_{max}$ controls the design domain volume via the Lagrangian, while $s_{max}$/$s_{min}$ bound the shape variables via a sigmoid.
>
> We have clarified our method by distinctly separating training from inference. These changes improve clarity and reproducibility.
>
> ***Table 1, Bolding*** Best metric values in Table 1 are bolded. Due to layout limitations, the table was split; we apologize for the confusion and will restore the original format.
>
> ***Statistical Significance*** We conducted a t-test on 20 independent runs, comparing each model to the best-performing model. The results are presented in the table below and and be provided in the revised appendix, confirming our choice of VAE architecture and training strategy based on $MSE_{sdf}$.
>
> |  | AE | MMD-VAE | VAE | AE | MMD-VAE | VAE | AE | MMD-VAE | VAE | AE | MMD-VAE | VAE |
> |---|---|---|---|---|---|---|---|---|---|---|---|---|
> | Strategy | st1 | st1 | st1 | st1 | st1 | st1 | st2 | st2 | st2 | st2 | st2 | st2 |
> | Decoder | Symm. | Symm. | Symm. | DeepSDF | DeepSDF | DeepSDF | Symm. | Symm. | Symm. | DeepSDF | DeepSDF | DeepSDF |
> | mean | 0.0014 | 0.00134 | 0.00128 | 0.000346 | 0.000368 | **0.000277** | 0.00155 | 0.0015 | 0.00146 | 0.000796 | 0.00059 | 0.000475 |
> | std | 0.000268 | 0.000233 | 0.000201 | 1.81e-05 | 1.95e-05 | **1.07e-05** | 0.000257 | 0.00023 | 0.000327 | 0.000101 | 5.42e-05 | 3.51e-05 |
> | p-value | 1.4e-11 | 3.9e-12 | 9.4e-13 | 4.9e-13 | 8.3e-15 | -- | 1.1e-12 | 3.3e-13 | 1.2e-10 | 4.1e-13 | 2.9e-14 | 8.8e-15 |
>
> Table: Metric: $MSE_{sdf}$ (p-values computed relative to st1\_VAE\_DeepSDF)
>
> In the case of Table 2, which presents the comparison results with other methods, we did not perform a t-test because these methods are deterministic with respect to the input parameters.
>
> ***Notations*** We apologize for the missing definitions. Inconsistencies in the method description have been corrected, and an updated version—including details on the shape code $\chi$—will appear in the main text, as well as a revised Figure 4. Additionally, a table of notations will be added to the appendix.

---

> > ### Comment · Reviewer_rbNQ · 2025-04-04
> >
> > I appreciate the experiments on free-form comparison. I understand that authors propose FMTO, but comparison to free-form is important to understand why FMTO is a better alternative. While NTopo is a free-form-based experiment, I believe it is not a published work. As I mentioned in the review, the authors do discuss diffusion-based topology optimization by Maze & Ahmed AAAI (Diffusion Models Beat GANs on Topology Optimization) but do not compare against it. Could the authors provide a discussion on why it cannot be compared to? Their work does release the code.

---

> > > ### Author Response · Authors · 2025-04-06
> > >
> > > We sincerely thank the Reviewer for their thoughtful feedback regarding the comparison with diffusion-based topology optimization, as presented in Mazé & Ahmed’s work. We appreciate your suggestion to discuss why a direct comparison with their approach, specifically the TopoDiff implementation, was not initially included, and we are pleased to provide further clarification.
> > >
> > > To explore this, we utilized the official TopoDiff implementation and adapted three test cases — MBB Beam Half, Cantilever Beam, and Beam Distributed Load — into a square domain, as TopoDiff’s implementation is designed to support square domains. We maintained the boundary conditions and load configurations as closely as possible. Below, we present the results for the three examples:
> > >
> > >
> > > | Method             | Method type | $\text{vonMises}_{max}$ | Compliance       | vf |
> > > | :---               | :---:       | :---:                   | :---:            | :---: |
> > > | SIMP               | Free-form   | $\textbf{132}$          | $\textbf{22.1}$  | 0.41 |
> > > | TopoDiff           | Free-form   | 133 $\pm$ 5.45          | 24.4 $\pm$ 0.602 | 0.415 $\pm$ 0.00417 |
> > > | NeuralHeavisideSDF | FMTO        | 132                     | 22.9             | 0.409 |
> > >
> > > Table: Methods comparison for Example 5: Square beam
> > >
> > > | Method             | Method type | $\text{vonMises}_{max}$      | Compliance                  | vf |
> > > | :---               | :---:       | :---:                        | :---:                       | :---: |
> > > | SIMP               | Free-form   | $\textbf{15.1}$              | $\textbf{4.09}$             | 0.41 |
> > > | TopoDiff           | Free-form   | (min = 65.2, max = 9.74e+04) | (min = 7.18, max = 1.5e+04) | 0.422 $\pm$ 0.006 |
> > > | NeuralHeavisideSDF | FMTO        | 18.3                         | 4.94                        | 0.411 |
> > >
> > > Table: Methods comparison for Example 6: Square beam Distributed Load
> > >
> > > | Method             | Method type | $\text{vonMises}_{max}$ | Compliance      | vf |
> > > | :---               | :---:       | :---:                   | :---:           | :---: |
> > > | SIMP               | Free-form   | $\textbf{28.8}$         | $\textbf{15.1}$ | 0.41 |
> > > | TopoDiff           | Free-form   | 64.5 $\pm$ 22.7         | 17 $\pm$ 0.368  | 0.424 $\pm$ 0.0048 |
> > > | NeuralHeavisideSDF | FMTO        | 30.2                    | 15.6            | 0.408 |
> > >
> > > Table: Methods comparison for Example 7: Square cantilever beam
> > >
> > > We observed that TopoDiff’s implementation is optimized for square domains and processes loads applied to a single node. In contrast, for cases such as Example 6 ("Square Beam Distributed Load"), where the load is distributed across multiple nodes, there is significant variability in the values of $\text{vonMises}_{max}$ and Compliance produced by TopoDiff’s framework.
> > >
> > > Our FMTO-based approach (NeuralHeavisideSDF) demonstrates competitive performance, particularly in maintaining lower Compliance values with lower material usage (vf) compared to TopoDiff in these adapted test cases. However, a direct comparison is challenging due to the differences in domain flexibility and load-handling capabilities. TopoDiff, as a free-form method, excels in optimizing within its square-domain, single-node-load paradigm, while our FMTO method prioritizes manufacturability by constraining solutions to predefined geometric primitives, which inherently limits its topological freedom compared to free-form approaches like TopoDiff or SIMP. This constraint inherently reduces topological flexibility compared to free-form methods like TopoDiff or SIMP, but it aligns with our goal of ensuring practical, manufacturable solutions. In the revision of the manuscript, we will include the discussion about TopoDiff and the above results in the revised manuscript.
> > >
> > > We have corrected the citation error that occurred when citing the discussed work and will ensure proper referencing in the revised manuscript. Thank you again for your constructive feedback, which has helped improve our paper. We hope that this response adequately addresses your concerns.

---

### Decision · Program_Chairs · 2025-05-01

**Decision:**

Accept (poster)

**Comment:**

This paper proposes a new method to simulate Feature-Mapping Topology Optimization (FMTO). Through incorporating a VAE-based heaviside function decoder, the method achieves superior practical solution compared with existing FMTO-type methods. Most reviewers are positive about the paper. Although some concerns were raised regarding novelty of the approach, the introduced techniques seem to be well-tailored for the application in a nontrivial way, thus could be considered a valid method contribution. Different concerns regarding writing, experiments, comparison with free-form methods, approximation of fine-detailed boundaries were raised, but were addressed satisfactorily during the rebuttal.